# Explain in Your Own Words: Improving Reasoning via Token-Selective Dual Knowledge Distillation

**Minsang Kim**
Korea University
SK Telecom
kmswin1@korea.ac.kr

**Seung Jun Baek**[*]
Korea University
sjbaek@korea.ac.kr

## Abstract

Knowledge Distillation (KD) can transfer the reasoning abilities of large models to smaller ones, which can reduce the costs to generate Chain-of-Thoughts for reasoning tasks. KD methods typically ask the student to mimic the teacher's distribution over the entire output. However, a student with limited capacity can be overwhelmed by such extensive supervision causing a distribution mismatch, especially in complex reasoning tasks. We propose Token-Selective Dual Knowledge Distillation (TSD-KD), a framework for student-centric distillation. TSD-KD focuses on distilling important tokens for reasoning and encourages the student to explain reasoning in its own words. TSD-KD combines indirect and direct distillation. *Indirect* distillation uses a weak form of feedback based on preference ranking. The student proposes candidate responses generated on its own; the teacher re-ranks those candidates as indirect feedback without enforcing its entire distribution. *Direct* distillation uses distribution matching; however, it selectively distills tokens based on the relative confidence between teacher and student. Finally, we add *entropy regularization* to maintain the student's confidence during distillation. Overall, our method provides the student with targeted and indirect feedback to support its own reasoning process and to facilitate self-improvement. The experiments show the state-of-the-art performance of TSD-KD on 10 challenging reasoning benchmarks, outperforming the baseline and runner-up in accuracy by up to 54.4% and 40.3%, respectively. Notably, a student trained by TSD-KD even outperformed its own teacher model in four cases by up to 20.3%. The source code is available at https://github.com/kmswin1/TSD-KD.

## 1 Introduction

Knowledge distillation (KD) (Buciluǎ et al., 2006; Hinton et al., 2015) is a technique to compress large "teacher" models into smaller "student" models. Especially for reasoning tasks, model compression can reduce the inference costs of generating long Chains-of-Thoughts (Wei et al., 2022; Snell et al., 2024; Yang et al., 2025). The *off-policy* KD trains a student on outputs generated by a teacher by matching logits (Hinton et al., 2015) or high-probability tokens (Kim & Rush, 2016). A drawback of off-policy KD is the distribution mismatch between teacher-generated data and the student's own output, which can harm generalization (Agarwal et al., 2024).

Recently, *on-policy* KD (Gu et al., 2023; Agarwal et al., 2024) was proposed in which the student is trained on its own generated output to mitigate distribution shift. A penalty such as KL-divergence is used to impose the teacher's feedback on the student's output. However, the methods are teacher-forcing, i.e., forcing the student to match the teacher's distribution on *every* generated token. A student with limited capacity can be overwhelmed by such extensive feedback. Furthermore, the level of reasoning skills to solve complex tasks can differ significantly between teacher and student. For example, problem-solving processes by a college senior and a middle school student can inherently differ. Thus, a direct KL-type penalty may cause a distribution mismatch. We consider *targeted* and

---

[*]Corresponding Author

*indirect* distillation; by targeted, we mean focusing on distilling "important" tokens only; by indirect, we mean subtle feedback to match the student's level of reasoning, giving room for self-improvement.

A metric for token importance is its entropy measured on the LLM's output distribution over vocabulary at the token position. High entropy means the model is uncertain about its output.

Recent work in RL (Wang et al., 2025) proposes policy updates with rewards on a small set of high-entropy tokens to improve reasoning performance. Those tokens act as important branching points in the reasoning process. We also measured the token entropy of student models and plotted it over token positions (Fig. 1). Two things are observed: 1) Some tokens have indeed significantly higher entropy than others. Many peaks are several times higher than the mean. 2) High-entropy tokens appear *early* in the reasoning traces. This motivates the design of a KD method for reasoning tasks that capture token entropy and its dependence on token positions.

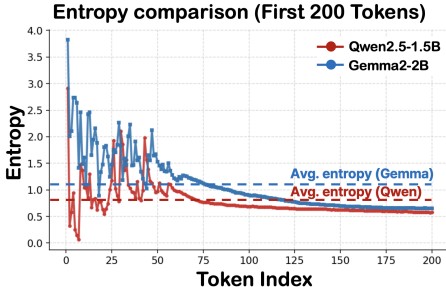

Figure 1: Average per token entropy of reasoning traces for (Yuan et al., 2025).

**Contribution.** We propose Token-Selective Dual Knowledge Distillation (TSD-KD), a student-centric framework for on-policy KD. TSD-KD selectively applies supervision to important tokens and encourages the student to reason in its own words. TSD-KD takes a dual approach combining indirect and direct distillation, both performed in a token-selective manner. In *indirect distillation*, the student proposes candidate tokens on its own at each reasoning step. The teacher gives only *preference* feedback on those candidates, instead of enforcing its entire output distribution as in traditional KD. This weak form of feedback helps the student self-improve and reduces distribution shift. We limit the distillation to the early part of reasoning (Fig. 1), and let the student freely generate reasoning for the rest. In *direct distillation*, we use KL-divergence type feedback which, however, is targeted at tokens the student is relatively uncertain. The distillation focuses on tokens that the student considers difficult, but the teacher considers easy. Finally, *entropy regularization* is added to the dual distillation. The token entropy is selectively *minimized* to reduce student's uncertainty and to boost its confidence. It acts as a regularization that maintains the student's confidence under distillation. Our key principle is to improve the reasoning ability of the student by strengthening its own reasoning process with minimal intervention from the teacher. Experiments on 10 reasoning benchmarks show the state-of-the-art performance of TSD-KD by up to 54.4% from the baseline student and up to 40.3% improvement from the second-best baseline. Notably, our student model even surpasses its teacher in four cases of reasoning tasks by up to 20.3%.

## 2 PRELIMINARIES

**On-policy Knowledge Distillation (KD).** In *on-policy* KD, the student model generates its own outputs, and the teacher model provides a supervisory signal on them. The student's output may improve as distillation progresses. It is distinguished from traditional *off-policy* distillation, which relies on fixed data pre-generated by the teacher. Since the student model uses its own generated outputs for training, the supervisory signal well-fits the student's distribution. This mitigates the distribution shift in off-policy KD and improves generalization performance (Agarwal et al., 2024).

**Model Definition.** The supervision can be implemented using metrics that encourage the student's output distribution to match that of the teacher. We first define the distribution $p$ of autoregressive language models given input $x$ and output response $y$. Suppose $y$ consists of $L$ tokens denoted by $y = (y_1, y_2, ..., y_L)$. The model distribution $p$ at token position $t = 1, \ldots, L$, is the softmax output of logits, i.e., $p(y_t|x, y_{<t}) := \exp(z_t)/\sum_{\hat{z}_t} \exp(\hat{z}_t)$ where $z_t \in \mathbb{R}$ is the output logit at position $t$.

**(Generalized) KL Divergence.** KL divergence is a metric used for KD as follows. Suppose $P$ and $Q$ are the model distributions of teachers and students, respectively. The forward KL-divergence is given by $\mathcal{D}_{\mathrm{KL}}(P\|Q) = \sum P(\cdot) \log \frac{P(\cdot)}{Q(\cdot)}$. In addition, reverse KL is defined as $\mathcal{D}_{\mathrm{KL}}(Q\|P)$. The forward KL leads to mode-covering, and the reverse KL leads to mode-seeking behavior of the student (Agarwal et al., 2024) (also see Appendix A.11 for an illustration). The choice between forward and reverse KL presents a key trade-off in KD. The generalized Jensen-Shannon Divergence (JSD) (Huszár, 2015) can control this trade-off by interpolating between both behaviors. Specifically,

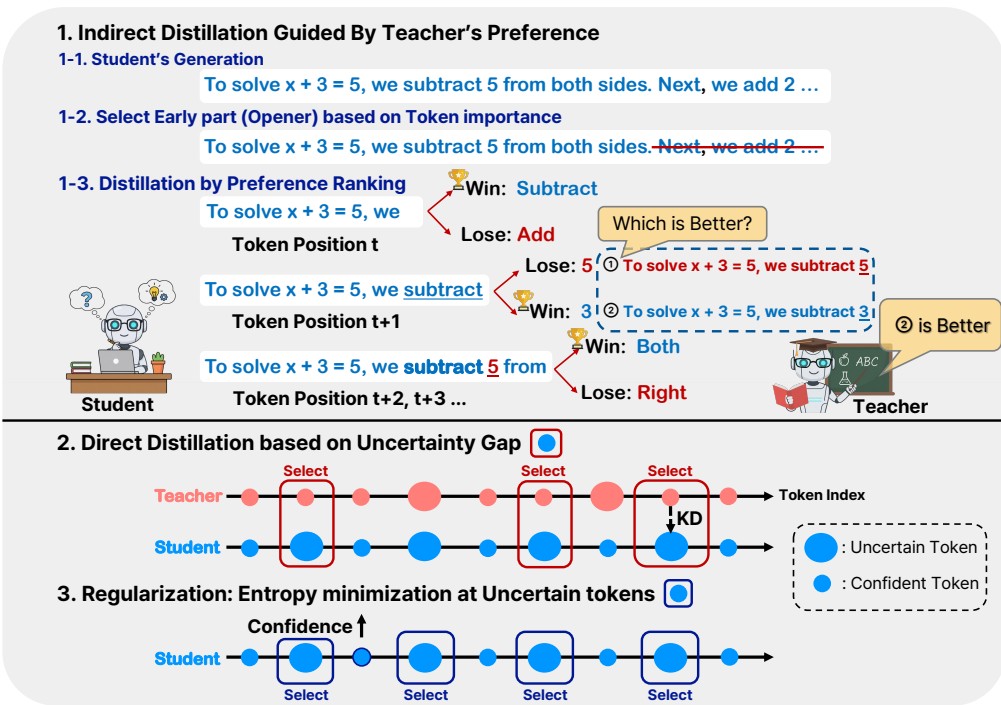

Figure 2: Overview of TSD-KD. **1) Indirect distillation.** Unlike traditional KD, the student actively suggests reasoning candidates, and the teacher chooses from them. **1.1)** The student generates a reasoning response. **1.2)** Early part of the response containing important branching of reasoning (called **opener**) is selected. **1.3)** The student sequentially generates candidates of partial responses from the opener. Top candidates are proposed to the teacher; for example, ① "To solve x+3=5, we subtract **5**" and ②"To solve x+3=5, we subtract **3**". The teacher provides preference ranking (prefers ②) for better reasoning. The preference signal is used as an indirect form of distillation. **2) Direct distillation** performs selective distillation of critical tokens about which the student is uncertain but the teacher is confident. **3) Entropy regularization** minimizes the entropy of critical tokens, reducing uncertainty and maintaining the student's confidence during distillation.

the divergence JSD($\beta$) for $0 < \beta < 1$ is defined as

$$\mathcal{D}_{\text{JSD}(\beta)}(P\|Q) = \beta \, \mathcal{D}_{\text{KL}}(P\|\beta P + (1-\beta)Q) + (1-\beta) \, \mathcal{D}_{\text{KL}}(Q\|\beta P + (1-\beta)Q). \quad (1)$$

It was shown that the gradient of JSD($\beta$) converges to that of forward KL when $\beta \to 0$ and reverse KL when $\beta \to 1$. In GKD (Agarwal et al., 2024), the following objective with JSD($\beta$) is minimized for the student to correct its errors in self-generated outputs. Denote the autoregressive models for the teacher by $p_T$ and the student by $p_S^\theta$ with learnable parameters $\theta$[1]. GKD minimizes the following:

$$\mathcal{D}\big(p_T\|p_S^\theta\big)(y|x) := \frac{1}{L}\sum_{t=1}^{L} \mathcal{D}_{\text{JSD}(\beta)}\big(p_T(\cdot|x, y_{<t})\|p_S^\theta(\cdot|x, y_{<t})\big). \quad (2)$$

The output response $y$ given input $x$ for this training can be sampled from either the teacher (off-policy) or the student (on-policy). As a result, the student will attempt to imitate the token-level distribution of the teacher throughout the output generation.

## 3 PROPOSED METHOD

**Outline.** The objective in (2) represents a *direct* distillation that forces the student to adjust its distribution to that of the teacher at every token. We advocate for being more student-centric and

---

[1]In case we make reference to the model $p_S^\theta$ with the gradient flow disabled, i.e., "stop-gradient", we will omit $\theta$ and simply use $p_S$.

introduce an *indirect* form of distillation that provides subtle feedback on students' generation. Our method will combine indirect and direct distillation, so that both work in a complementary manner. Importantly, the combined distillation is performed in a *token-selective* manner, focusing on tokens that the student considers difficult to predict. In contrast, we provide no or limited feedback for the rest of the tokens, allowing the student to freely generate its reasoning. The key idea is to encourage the student to explain reasoning in its own words to promote reasoning capability. In summary, we propose a student-centric approach combining indirect (Sec. 3.1) and direct distillation (Sec. 3.2) in addition to entropy regularization (Sec. 3.3), all of which are performed in a token-selective manner.

## 3.1 INDIRECT DISTILLATION GUIDED BY TEACHER'S PREFERENCE

**Motivation.** We introduce an indirect distillation guided by the teacher's preference. The idea is to leverage the teacher as a *ranking oracle* over the student's candidate predictions. At each token position, the student proposes its top-$k$ candidate tokens to the teacher. The teacher *re-ranks* these tokens, and the student aligns the candidate tokens with the teacher's ranking. Importantly, it is the student who proposes the candidate tokens; the teacher only re-ranks the student's proposal. Unlike traditional KD, the teacher's top choices (e.g., its own top-$k$) are unknown to the student. Thus, this is an indirect form of distillation. In the following, we describe the detailed algorithm.

**Distillation by Preference Ranking.** Consider the output at token position $t$ given input $x$. It is the next token predicted by the student given $x$ and $y_{<t}$. Let $z_t[1], z_t[2], \ldots, z_t[k]$ denote the *logits* of $k$-best candidate tokens predicted by the student in the descending order. Since $z_t$'s are logits, we have $z_t[1] \geq z_t[2] \geq \ldots \geq z_t[k]$. The teacher's ranking of these $k$ candidates is denoted by permutation $\pi_t : \{1, \ldots, k\} \rightarrow \{1, \ldots, k\}$. Thus, $z_t[\pi_t(i)]$ is the student's logit of the $i$-th best choice by the teacher. The teacher ranks the tokens by its own logits. The teacher first computes its logits at position $t$ given $x$ and $y_{<t}$. The teacher then produces the ordering $\pi_t$ by ranking its logits corresponding to the student's candidate tokens.

For the preference modeling, we adopt the Plackett-Luce (PL) model (Plackett, 1975) widely used for aligning LLMs with human preference (Ouyang et al., 2022; Rafailov et al., 2023). The PL framework models the probability of the ranking of items. Suppose there are $k$ items indexed by $1, \ldots, k$ and the $i$-th item is associated with a reward (score) $r(i)$. The probability that a user prefers (ranks) the items in the order of a permutation $\pi$ of $\{1, \ldots, k\}$ is given by

$$\prod_{j=1}^{k} \frac{\exp(r(\pi(j)))}{\sum_{\ell=j}^{k} \exp(r(\pi(\ell)))} \qquad \text{for } k = 2, 3, 4, \ldots \tag{3}$$

where case $k = 2$ is the Bradley-Terry (BT) model (Bradley & Terry, 1952). We use the PL model to formulate the probability that the student's token ranking is aligned with that of the teacher's. The likelihood that students' candidates are ranked according to $\pi_t$, denoted by $P_{\text{PL}}(\pi_t \mid x, y_{<t})$, is

$$P_{\text{PL}}(\pi_t \mid x, y_{<t}) = \prod_{j=1}^{k} \frac{\exp(z_t[\pi_t(j)])}{\sum_{\ell=j}^{k} \exp(z_t[\pi_t(\ell)])}. \tag{4}$$

In (4), we directly use logits $z_t$ as the score for the PL distribution. Its implications will be explained in the next paragraph. We minimize its negative log-likelihood over token positions:

$$\mathcal{L}_{\text{Indirect}} = -\sum_{t} \log P_{\text{PL}}(\pi_t \mid x, y_{<t}). \tag{5}$$

Recent work on on-policy KD (Gu et al., 2023; Agarwal et al., 2024) is analogous to imitation learning (Ross et al., 2011) where a student simply mimics the output distribution of the teacher. In contrast, the student in our model learns to reason over its own generated choices. This self-reflective nature not only mitigates distribution shift, but also enables improved generalization and reasoning, especially for complex tasks.

**Learning Preferences of Sub-responses.** We explain the implication of (4) in the context of preference modeling. Given the response $y$, we define the *sub-response* of length $t$ as the partial response consisting of the beginning $t$ tokens of $y$ denoted by $y_{\rightarrow t} := (y_1, y_2, \ldots, y_t)$. Thus, $y_{\rightarrow t}$ is the partial response that represents the reasoning trace of reaching the token at position $t$. We show

that distribution (4) is a ranking model for such partial responses as follows. For simplicity, let $k = 2$ and the student's top-2 tokens at position $t$ be $y_t^{(1)}, y_t^{(2)}$. Consider two sub-responses:

$$y_{\to t}^{(1)} = (y_1, y_2, ..., y_{t-1}, y_t^{(1)}), \quad y_{\to t}^{(2)} = (y_1, y_2, ..., y_{t-1}, y_t^{(2)}) \tag{6}$$

That is, the sub-responses differ only at position $t$. We show that, under some choices of reward functions, (4) is the distribution of ranking of candidate sub-responses $y_{\to t}^{(1)}$ and $y_{\to t}^{(2)}$. For example, consider two sub-responses (Fig. 2): $y_{\to t}^{(1)} =$"To solve x+3=5, we subtract **3**", and $y_{\to t}^{(2)} =$"To solve x+3=5, we subtract **5**". The teacher chooses the preferred sub-response for better reasoning, and our model aligns the student with the choice. The following proposition shows our claim for $k = 2$ for simplicity (BT model (Bradley & Terry, 1952)). It is straightforward to extend the claim for $k > 2$ (PL model (Plackett, 1975)).

**Proposition 1.** *Suppose the student's reward for response $y$ given input $x$ is implicitly defined as*

$$r_S(x, y) := \log p_S(y|x) = \sum_t \log p_S(y_t|y_{<t}, x) \tag{7}$$

*Similarly define the teacher's reward function $r_T(x, y) := \log p_T(y|x)$. Suppose top-2 tokens at position $t$ are $y_t^{(1)}, y_t^{(2)}$, and sub-responses up to $t$ are given as in (6). Consider the preference model where **teacher's reward** determines the preference label for $y_{\to t}^{(1)}$ and $y_{\to t}^{(2)}$. That is, $y_{\to t}^{(1)}$ is preferred over $y_{\to t}^{(2)}$ given $x$ (denoted by $y_{\to t}^{(1)} \succ y_{\to t}^{(2)}|x$) if*

$$r_T(x, y_{\to t}^{(1)}) \geq r_T(x, y_{\to t}^{(2)}) \tag{8}$$

*and vice versa. **Student's reward** determines the preference distribution: the probability that $y_{\to t}^{(1)}$ is preferred over $y_{\to t}^{(2)}$ is given by*

$$p(y_{\to t}^{(1)} \succ y_{\to t}^{(2)}|x) = \frac{\exp(r_S(x, y_{\to t}^{(1)}))}{\exp(r_S(x, y_{\to t}^{(1)})) + \exp(r_S(x, y_{\to t}^{(2)}))} \tag{9}$$

*Then (4) is equivalent to (9) for $k = 2$. Also, $\mathcal{L}_{Indirect}$ is the negative log-likelihood of PL distribution (4) aligning the student's and teacher's preferences on sub-responses $y_{\to t}$ summed over all $t$.*

See Appendix A.1 for proof. Another implication of Proposition 1 is that the preference distribution at token-level (4) is equivalent to that at sentence-level (9), under the preference reward (7) (preference rewards based on log-model probability are commonly used, e.g., (Rafailov et al., 2023)). This shows that our loss $\mathcal{L}_{Indirect}$ has significant computational benefits. $\mathcal{L}_{Indirect}$ in (5) computes a running sequence of word (token)-level losses; however, this turns out to be a running sequence of sentence-level losses under reward (7). Considering that a sentence is a cumulative sequence of words, our loss enables much efficient comparison of partial sentences by reducing it down to word-level comparison.

**Token-selective Distillation: Importance of Openers.** Although we use an indirect form of distillation, it may still drift the student's distribution towards that of teachers. We hypothesized that it is desirable to provide preference guidance on difficult tokens but not on the rest of the tokens, so that the student freely chooses its own words for the remaining process of reasoning. As shown in Fig. 1, difficult tokens are concentrated on the beginning part of responses. This is reasonable; it is crucial to set the initial direction of reasoning correctly, as it establishes the initial branching of reasoning traces. Thus, we focus on the high-entropy tokens in the initial part of responses.

We introduce the concept of *opener*: it is the consecutive token sequence in the beginning part of

---

**Algorithm 1** Indirect Distillation by Teacher-Guided Preference Ranking

1: **Given**: Teacher model $p_T$, Student Model $p_S^\theta$, Prompt dataset $X$
2: Sample $x$ from $X$, generate $y \sim p_S^\theta(\cdot|x)$
3: Select the *opener* of $y$ up to top-$c\%$ cumulative entropy
4: **for** each token position $t$ in *opener* **do**
5:     Select top-$k$ logits $\{z_t[1], ..., z_t[k]\}$ by student $p_S^\theta(\cdot|y_{<t}, x)$
6:     Re-rank logits to get the order of preference $\pi_t$ by teacher $p_T(\cdot|y_{<t}, x)$
7:     Compute $P_{\text{PL}}(\pi_t \mid x, y_{<t})$
8: **end for**
9: Compute $\mathcal{L}_{\text{PL}} = -\sum_t \log P_{\text{PL}}(\pi_t \mid x, y_{<t})$

---

a response where *cumulative entropy first reaches* $c\%$. Specifically, for an autoregressive model $p$, the entropy of response $y$ at position $t$ is defined as

$$H_t(p) = -\sum_{v \in \mathcal{V}} p(v \mid x, y_{<t}) \log p(v \mid x, y_{<t}) \tag{10}$$

where $\mathcal{V}$ denotes the vocabulary (we omit the dependence of $H_t(\cdot)$ on $x$ and $y$ for notational simplicity). The opener of response $y$ of length $L$ is the initial token sequence up to position $m$ where $m$ is the minimum integer satisfying $\sum_{t=1}^{m} H_t(p_S) / \sum_{t=1}^{L} H_t(p_S) \geq c\%$.

In this work, we use a small value of $c = 10\%$ which proves to be sufficient (we provide a study on other values of $c$ in Sec. 4). In addition, we can make $c$ *adaptive* where $c$ is determined based on the estimated difficulty of the sample. If the sample difficulty is estimated to be high, $c$ is set to a larger value, i.e., a higher fraction of the sample is distilled to the student model. The detailed method of adaptation and related experiments are provided in Appendix A.7. Finally, our indirect distillation is performed in a token-selective manner, i.e., only on the tokens in the opener. A pseudocode is shown in Algorithm 1.

## 3.2 DIRECT DISTILLATION BASED ON UNCERTAINTY GAP

Our indirect distillation is a student-centric approach to refining the reasoning path early in response generation. However, if the input question is very challenging, the student's top candidate tokens at reasoning steps may be off from the correct path. To address this issue, we use direct distillation with the JSD objective in (2). However, we apply direct distillation in a *token-selective* manner. Specifically, the distillation focuses on tokens for which the student is highly uncertain (high entropy), but the teacher is highly confident (low entropy). Thus, the distillation targets tokens that exhibit large uncertainty gaps. We implement the selective mechanism using a *gating function* as follows.

$$\mathcal{L}_{\text{Direct}} = \frac{1}{L} \sum_{t=1}^{L} \underbrace{\sigma_\tau(H_t(p_S) - H_t(p_T))}_{\text{gating function}} \cdot \mathcal{D}_{\text{JSD}(\beta)}(p_T(\cdot|x, y_{<t}) || p_S^\theta(\cdot|x, y_{<t})), \tag{11}$$

where we use sigmoidal gating $\sigma_\tau(u) := (1 + \exp(-u/\tau))^{-1}$, and $\tau > 0$ controls the sharpness of the entropy gap. Thus, $\sigma_\tau$ dynamically modulates the strength of the distillation: we reinforce the distillation on those tokens with large $H_t(p_S) - H_t(p_T)$, i.e., the student is uncertain while the teacher is highly confident. Moreover, the gating mechanism tries *not* to distill tokens in which the student is relatively confident. This helps the student generate its reasoning freely. Thus, our distillation is student-centric and also reduces the distribution shift. Our gating mechanism can be considered as a *soft selection* of tokens based on the relative confidence between student and teacher. Moreover, the soft selection of tokens has an effect of scaling the gradient of the student model during training, which can accelerate the convergence. We provide a related analysis in the Appendix A.2.

We comment on another complementary aspect of indirect and direct distillation. Indirect distillation provides sequence-level guidance in setting up the early reasoning path. Direct distillation provides token-level and targeted guidance in precisely generating important tokens. Thus, the combination of indirect and direct distillation has synergistic effects on student-centric learning.

## 3.3 ENTROPY REGULARIZATION AND FINAL LOSS

**Boosting Confidence with Entropy Regularization.** Recent findings show that *minimizing* the token entropy, i.e., making the probability of high-confident output even higher, improves the reasoning capability of LLMs (Agarwal et al., 2025). It reduces uncertainty in intermediate steps, leading to more reliable reasoning trajectories. Inspired by this approach, we perform the entropy minimization, however, in a *token-selective* manner. We minimize the entropy of the most uncertain tokens only. That is, the student is guided to boost confidence on critical reasoning tokens only. However, there is no feedback for the remaining tokens, allowing the student to freely generate reasoning, similar to our distillation methods. Formally, let $\mathcal{I} \subseteq \{1, \ldots, T\}$ denote the index set of the top-10% tokens with the largest entropy values $\{H_t(p_S^\theta)\}_{t=1}^{T}$. We minimize the following objective:

$$\mathcal{L}_{\text{EM}} = \mathbb{E}_{x \sim \mathcal{X}} \left[ \frac{1}{|\mathcal{I}|} \sum_{t \in \mathcal{I}} H_t(p_S^\theta) \right] \tag{12}$$

The choice of top-10% is motivated by the ratio of important tokens suggested for RL for reasoning (Wang et al., 2025). They showed that policy updates from tokens in the top 10–20% of the entropy values improve performance. For completeness, we provide study on varying ratios in Appendix A.4.

The entropy minimization has a *regularization* effect. Similar to indirect and direct distillation, the focus is on difficult tokens. For these tokens, the student is made more deterministic (confident) rather than explorative through entropy minimization. This can improve the effectiveness of distilling challenging knowledge.

**Final Loss.** The overall training objective is

$$\min_{\theta} \; \mathbb{E}_{x \sim \mathcal{X}} \left[ \alpha \mathcal{L}_{\text{Indirect}} \; + \mathcal{L}_{\text{Direct}} \; + \mathcal{L}_{\text{EM}} \right] \tag{13}$$

where $\alpha > 0$ controls the relative weight of the indirect distillation. For simplicity, we set the relative weights of direct distillation and entropy minimization equal because they both pertain to entropy.

## 4 EXPERIMENT

### 4.1 EXPERIMENTAL SETTINGS

**Overview.** In our experiments, we mainly compare on-policy distillation methods. We train our method (TSD-KD) and recent state-of-the-art baselines using on-policy generations from the student model. For completeness, we also evaluated off-policy baselines. For some hyperparameters, we used $k = 10$ for indirect distillation. For direct distillation, we set $\beta = 0.9$ for $\text{JSD}(\beta)$ as suggested by GKD (Agarwal et al., 2024). Detailed hyperparameters are provided in Appendix A.9.

**Models.** We conduct our main experiments with the Qwen2.5-Instruct model family (Qwen et al., 2024), using the 14B parameter model as the teacher and the 1.5B model as the student. To demonstrate generalizability, we also evaluate our approach on Gemma2 (Team et al., 2024), with the 9B and 2B models serving as teacher and student, respectively.

**Baselines.** We compare our method with baselines in two categories. **On-policy methods:** We benchmark against recent on-policy approaches, including DistilLLM (Ko et al., 2024), MiniLLM (Gu et al., 2023), GKD (Agarwal et al., 2024) with $\beta = 0.9$, and Speculative KD (Xu et al., 2025). These methods are trained on the outputs generated from the student. **Off-policy methods:** We include traditional KD approaches such as Supervised-KD (Hinton et al., 2015) and Sequence-level KD (Kim & Rush, 2016). Under these methods, the student is trained on a static dataset of responses generated by the teacher.

**Training Data.** For on-policy methods, we use prompts from UltraInteract (Yuan et al., 2025) and let the student generate responses to the prompts. The responses are used as training data. For off-policy methods, we use the teacher's responses to the same prompts as training data.

**Evaluation.** We assess the models on a diverse set of reasoning benchmarks. This includes *Mathematical reasoning* (GSM8K (Cobbe et al., 2021), GSM-Plus (Li et al., 2024), MATH (Hendrycks et al., 2021b), MMLU-Pro-Math (Wang et al., 2024)), *STEM and scientific reasoning* (MMLU-STEM (Hendrycks et al., 2021a), ScienceQA (SciQ) (Welbl et al., 2017)), *Program synthesis* (MBPP (Austin et al., 2021)), *Broad reasoning* (BBH (Suzgun et al., 2022), MuSR (Sprague et al., 2024)), and *Instruction following* (IFEval (Zhou et al., 2023)). For all benchmarks, we report accuracy following the standard evaluation protocols. More details on the benchmarks are in Appendix A.10.

### 4.2 RESULTS

**Main Results.** As presented in Table 1, TSD-KD achieves the best performance across six tasks. TSD-KD significantly outperforms all the baseline methods. The margin of improvement is up to 54.4% over the baseline student. The largest gain is achieved in the challenging MATH benchmark; TSD-KD (26.1) surpasses the second-best method by 40.3%. Surprisingly, it even outperforms its own teacher (21.7) by 20.3%. These results indicate that TSD-KD enables more generalizable reasoning as compared to simply mimicking the teacher's distribution.

We evaluated TSD-KD on more challenging reasoning benchmarks, as shown in Table 2. TSD-KD consistently achieves the best performances across all four challenging tasks, where TSD-KD leads

| Method | GSM8K | GSM-Plus | MATH | MBPP | IFEval | MMLU-STEM |
|---|---|---|---|---|---|---|
| Qwen2.5-14B (Teacher) | 80.3 | 59.7 | 21.7 | 78.9 | 85.9 | 70.5 |
| Qwen2.5-1.5B (Student) | 57.1 | 38.8 | 16.9 | 38.4 | 53.1 | 49.5 |
| Sequence-Level KD | 56.5 | 38.2 | 16.5 | 40.6 | 53.7 | 48.9 |
| Supervised-KD | 56.3 | 37.8 | 17.9 | 41.0 | 51.4 | 48.4 |
| DistiLLM | 57.2 | 38.7 | 18.6 | 42.1 | 52.5 | 47.8 |
| Speculative KD | 58.0 | 39.2 | 18.5 | 41.6 | 52.4 | 47.8 |
| MiniLLM | 57.7 | 39.7 | 17.8 | **42.2** | 54.7 | 48.2 |
| GKD ($\beta = 0.9$) | 57.9 | 39.9 | 18.1 | 41.8 | 52.3 | 47.7 |
| TSD-KD | **60.1** | **40.5** | **26.1*** | 42.1 | **55.2** | **50.0** |

Table 1: Performance comparison between distillation methods in Qwen2.5 (14B → 1.5B). Best performance in **Bold** font. (*) mark for the student model outperforming the teacher.

| Method | MMLU-Pro-Math | SciQ | BBH | MuSR |
|---|---|---|---|---|
| Qwen2.5-14B (Teacher) | 77.6 | 86.4 | 60.8 | 60.8 |
| Qwen2.5-1.5B (Student) | 34.4 | 85.5 | 36.5 | 38.7 |
| Sequence-Level KD | 32.6 | 87.9 | 36.6 | 36.6 |
| Supervised-KD | 35.3 | 86.6 | 36.4 | 37.9 |
| DistiLLM | 35.2 | 85.7 | 36.5 | 38.7 |
| Speculative KD | 34.8 | 86.7 | 37.2 | 37.7 |
| MiniLLM | 36.2 | 84.1 | 36.0 | 38.3 |
| GKD ($\beta = 0.9$) | 35.7 | 86.6 | 36.2 | 38.5 |
| TSD-KD | **36.9** | **93.0*** | **40.2** | **39.6** |

Table 2: Performance comparison of more advanced benchmarks in Qwen2.5 (14B → 1.5B). Best performance in **Bold** font. (*) mark for the student model outperforming the teacher.

| $\sigma_\tau(\cdot)$ | $\mathcal{L}_{\text{Indirect}}$ | $\mathcal{L}_{\text{EM}}$ | GSM8K | GSM-Plus | MATH | IFval | MMLU-STEM |
|---|---|---|---|---|---|---|---|
| ✗ | ✗ | ✗ | 57.9 | 39.9 | 18.1 | 52.3 | 47.7 |
| ✔ | ✗ | ✗ | 58.3 | 40.3 | 18.2 | 54.2 | 48.1 |
| ✔ | ✔ | ✗ | 58.6 | 40.4 | 20.9 | 54.9 | 49.4 |
| ✔ | ✗ | ✔ | 59.0 | 39.8 | 22.3 | 54.1 | 49.3 |
| ✔ | ✔ | ✔ | **60.1** | **40.5** | **26.1** | **55.2** | **50.0** |

Table 3: Ablation study of each component in Qwen2.5 (14B → 1.5B). The baseline is on-policy JSD with a $\beta = 0.9$ model (except all components).

the runner-up by 9.8%. Remarkably, on complex scientific reasoning tasks in SciQ, the student model trained with TSD-KD surpasses its teacher by a significant margin of 7.6%. The results show the effectiveness of our student-centric approach to distilling knowledge in pivotal reasoning steps.

**Ablation Study.** Table 3 presents an ablation study on three main components of TSD-KD. The first column represents the use of token-selective gating $\sigma_\tau(\cdot)$ in direct distillation. The second column represents the use of indirect distillation ($\mathcal{L}_{\text{Indirect}}$). The third column represents the use of selective entropy minimization ($\mathcal{L}_{\text{EM}}$). The default baseline without any of these components is the vanilla GKD with $\beta = 0.9$, which is the fully direct KD on all tokens.

The results show the performance gains achieved by adding each component. The component $\sigma_\tau(\cdot)$ yields an initial gain of up to 3.6%. Incorporating $\mathcal{L}_{\text{Indirect}}$ increases performance by up to 15.5%; it significantly improves performance on general reasoning tasks such as MMLU-STEM. Lastly, $\mathcal{L}_{\text{EM}}$ is effective for mathematical reasoning, increasing the MATH score from 18.1 to 22.3. The fully

| Method | GSM8K | GSM-Plus | MATH | MBPP | IFEval | MMLU-STEM |
|---|---|---|---|---|---|---|
| Gemma2-9B (Teacher) | 57.7 | 40.8 | 25.8 | 40.0 | 64.7 | 50.2 |
| Gemma2-2B (Student) | 53.7 | 36.2 | 16.1 | 39.2 | 62.6 | 25.5 |
| Sequence-Level KD | 53.4 | 33.8 | 17.7 | 39.2 | 62.4 | 25.9 |
| Supervised-KD | 53.6 | 36.8 | **17.9** | 39.0 | 63.4 | 25.5 |
| DistiLLM | 54.3 | 35.8 | 17.8 | 39.6 | 62.2 | 25.2 |
| MiniLLM | 54.4 | 36.8 | 17.5 | **40.6*** | 65.8 | 25.9 |
| GKD ($\beta = 0.9$) | 54.1 | 36.9 | 17.2 | 40.0 | 63.5 | 26.2 |
| TSD-KD | **55.3** | **37.9** | 18.9 | 40.4* | **66.5*** | **27.0** |

Table 4: Performance comparison between different distillation methods in Gemma2 (9B → 2B). Best performance in **Bold** font. (**\***) mark for the student model outperforming the teacher.

| Method | GSM8K | MATH | GSM_Plus | MBPP | IFEVAL | MMLU_STEM |
|---|---|---|---|---|---|---|
| Qwen3-8B (teacher) | 82.9 | 22.4 | 59.4 | 66.0 | 86.5 | 40.2 |
| Qwen3-1.7B (student) | 66.3 | 21.0 | 43.1 | 50.8 | 75.1 | 36.2 |
| Sequence KD | 65.7 | 20.8 | 43.5 | 51.3 | 75.7 | 36.9 |
| Supervised KD | 66.5 | 21.5 | 43.7 | 51.2 | 76.1 | 36.5 |
| DistiLLM | 66.7 | 21.2 | 45.5 | 51.4 | 78.2 | 36.8 |
| MiniLLM | 67.0 | 22.6 | 46.4 | **52.1** | 79.2 | 36.5 |
| GKD | 67.2 | 22.8 | 46.0 | 51.7 | 79.1 | 36.7 |
| TSD-KD | **68.7** | **28.0*** | **47.2** | 51.9 | **79.8** | **37.5** |

Table 5: Performance comparison between different distillation methods in Qwen3 (8B → 1.7B). Best performance in **Bold** font. (**\***) mark for the student model outperforming the teacher.

integrated TSD-KD model achieves performance improvement up to 44.2% over the baseline. Our results indicate a strong synergistic effect among the key components of TSD-KD.

**Generalization to Gemma2 Models.** We evaluate our method on a different base model and extend our evaluation to Gemma2 (Team et al., 2024). We apply the same set of distillation methods to a Gemma2-9B teacher and a Gemma2-2B student. Table 4 shows that TSD-KD achieves the highest performance across 5 tasks. While some competing methods, such as GKD and MiniLLM, show strong performance on GSM8K and MBPP, TSD-KD performs the best on the majority of tasks, including MATH and IFEval.

Importantly, the Gemma2-2B student model trained by TSD-KD surpasses the Gemma2-9B teacher, on both the IFEVAL and MBPP benchmarks. This outcome of *student surpasses teacher*, similar to the findings for Qwen2.5, underscores the effectiveness of student-centric learning. The results of TSD-KD for Gemma2 validate its effectiveness as a general framework for distillation.

**Generalization to Qwen3 Models.** We further evaluate TSD-KD on Qwen3 LLMs, using the 8B model as the teacher and the 1.7B model as the student. As shown in Table 5, TSD-KD achieves state-of-the-art performance in most of the benchmarks. Moreover, on the challenging MATH benchmark, the student model from our method outperforms the teacher model by a large margin (**+25.0%**); this improvement is even greater than the case of Qwen 2.5 (+20.3%) on MATH. These results confirm that our method consistently improves reasoning performance across multiple benchmarks, demonstrating its effectiveness on more competitive reasoning models such as Qwen3-8B.

**Analysis of Openers in Indirect Distillation.** The opener is a key concept in our token-selective indirect distillation. The length of the opener is defined as the token position from the beginning until we reach $c\%$ of the cumulative entropy. There are two settings for the hyperparameter $c$: fixed or adaptive. We examine the effect of $c$ when we use fixed $c$.

Fig. 3 shows the average performance on the benchmarks from Table 1 with varying $c$. We observe that the peak performance is achieved at $c = 10\%$. This indicates that the teacher's feedback should focus on a relatively small portion early in the student's reasoning trajectory.

Interestingly, performance worsens at $c = 100\%$, which is equivalent to applying distillation to all tokens. This result strongly supports our key hypothesis: a strategic selection of tokens is essential. It is important to indirectly provide feedback on the initial direction of reasoning, keeping the teacher's intervention minimal.

**Additional experiments.** We provide additional experiment results on: impact of $k$ in top-$k$ token selection in indirect distillation (A.3); the ratio of token selection in entropy regularization (A.4); impact of $\beta$ in JSD($\beta$) for direct distillation (A.5); comparison of on-policy vs. off-policy (A.6); Adaptive selection of $c$ (A.7); and KD with PEFT (A.8).

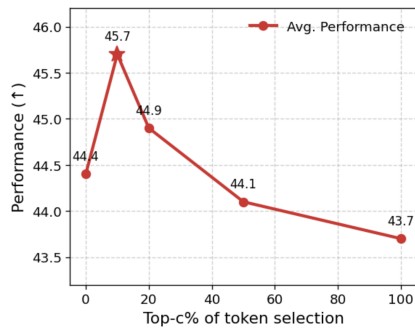

Figure 3: Performance with varying $c$.

## 5 RELATED WORK

**Knowledge Distillation of LLMs.** Knowledge Distillation (KD) was initially proposed as a method for a student model to imitate the output probability distribution of a teacher model (Buciluă et al., 2006; Hinton et al., 2015). This was later extended to Sequence-Level KD (Kim & Rush, 2016), which trains the student by maximizing the likelihood of high-probability sequences from the teacher. However, these approaches suffer from distribution shift, a discrepancy between the teacher-provided training distribution and the student's own generated distribution at inference. To mitigate this, recent research has shifted towards on-policy KD methods that use the student's own generations for training (Lin et al., 2020; Gu et al., 2023; Ko et al., 2024; Agarwal et al., 2024; Xu et al., 2025). KPOD (Feng et al., 2024) proposed a progressive off-policy KD that transfers the teacher's CoT reasoning to the student by selective distillation of key tokens combined with curriculum learning. In contrast, we take an on-policy, dual distillation approach with token selection based on entropy. Although these methods have shown success in general tasks, their application to distilling complex reasoning remains underexplored. Building on this paradigm, our work proposes a student-centric framework for on-policy distillation to enhance complex reasoning capabilities.

**Advanced Reasoning with LLMs.** Chain-of-Thought (CoT) prompting was introduced in (Kojima et al., 2022; Wei et al., 2022) for solving complex reasoning tasks by generating intermediate steps in a step-by-step manner. Following the success of CoT, more sophisticated reasoning strategies have been proposed. Tree-of-Thoughts (ToT) (Yao et al., 2023) proposed a multi-path CoT, exploring and evaluating multiple possible reasoning paths in a tree structure to find the optimal solution. Graph-of-Thoughts (GoT) (Besta et al., 2024) introduced the reasoning process as a graph structure, allowing for more flexible combinations of thoughts. The success of these advanced reasoning techniques is attributed to the enhanced reasoning capabilities of recent LLMs (Guo et al., 2025; Yang et al., 2025). However, a common limitation of these methods is the high computational cost required to generate the reasoning process. This issue has highlighted the importance of KD, which aims to transfer the reasoning abilities of these large, computationally expensive models to more efficient ones. The proposed KD method focuses on developing a model with advanced reasoning capabilities while maintaining computational efficiency.

## 6 CONCLUSION

We propose Token-Selective Dual Knowledge Distillation (TSD-KD) which enhances the reasoning performance of student models by focusing distillation on important tokens. TSD-KD employs indirect distillation, where the teacher provides a preference ranking for the student's own generated output. Direct distillation is also applied selectively to a targeted set of tokens based on the relative uncertainty between teacher and student. Finally, TSD-KD maintains the student's confidence with entropy regularization. Experimental results on various reasoning benchmarks show that TSD-KD substantially outperforms existing on-policy KD methods and even surpasses the teacher model in several cases. Our study demonstrates the effectiveness of student-centered approaches in KD.

## 7 ACKNOWLEDGEMENT

This work was partly supported by ICT Creative Consilience Program through the Institute of Information & Communications Technology Planning & Evaluation (IITP) grant funded by the Korea government (MSIT)(IITP-2026-RS-2020-II201819, 50%), and by the National Research Foundation of Korea (NRF) grant funded by the Korea government (MSIT) (RS-2022-NR070834).

## 8 REPRODUCIBILITY STATEMENT

**Training Implementation.** We implement our training framework based on TRL. This framework provides trainers with various loss functions, but we develop proposed loss functions.

**Inference Implementation.** We generate all generate datasets using vLLM.

**Evaluation Datasets.** We evaluate various math and code tasks in lm-evaluation-harness, which are open-access repositories.

**Source Code.** We uploaded the source code in the public repository.

**Computational Resources.** All our experiments, we used eight A100 GPUs with 80GB VRAM.

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

## A  APPENDIX

### A.1  PROOF OF PROPOSITION 1

Let $p_S(y|x)$ denote the autoregressive student model defined as follows:

$$p_S(y_t|y_{<t}, x) := \frac{\exp(z_t)}{\sum_{\hat{z}_t} \exp(\hat{z}_t)}$$

where $z_t$ is the logit of the output token at position $t$. This implies that, for some given sub-response $y_{\to t}$, we have that

$$r_S(x, y_{\to t}) = \sum_{s=1}^{t} \log p_S(y_s|y_{<s}, x) = \sum_{s=1}^{t} \left[ z_s + \log \left( \sum_{\hat{z}_s} \exp \hat{z}_s \right) \right]$$

Then, given the sub-responses $y_{\to t}^{(1)}$ and $y_{\to t}^{(2)}$, we have

$$r_S(x, y_{\to t}^{(1)}) = z_t^{(1)} + \underbrace{\log \left( \sum_{\hat{z}_t} \exp \hat{z}_t \right) + \sum_{s=1}^{t-1} \left[ z_s + \log \left( \sum_{\hat{z}_s} \exp \hat{z}_s \right) \right]}_{:=\mathcal{Z}(t)} \tag{14}$$

and

$$r_S(x, y_{\to t}^{(2)}) = z_t^{(2)} + \underbrace{\log \left( \sum_{\hat{z}_t} \exp \hat{z}_t \right) + \sum_{s=1}^{t-1} \left[ z_s + \log \left( \sum_{\hat{z}_s} \exp \hat{z}_s \right) \right]}_{:=\mathcal{Z}(t)} \tag{15}$$

Note that $\mathcal{Z}(t)$ is identical for two sub-responses $y_{\to t}^{(1)}$ and $y_{\to t}^{(2)}$, because they differ only in the last token at position $t$. Then, the PL distribution of preference with the student's reward $r_S$ defined in (9) is given by

$$p(y_{\to t}^{(1)} \succ y_{\to t}^{(2)}|x) = \frac{\exp(r_S(x, y_{\to t}^{(1)}))}{\exp(r_S(x, y_{\to t}^{(1)})) + \exp(r_S(x, y_{\to t}^{(2)}))} \tag{16}$$

$$= \frac{\exp \left[ z_t^{(1)} + \mathcal{Z}(t) \right]}{\exp \left[ z_t^{(1)} + \mathcal{Z}(t) \right] + \exp \left[ z_t^{(2)} + \mathcal{Z}(t) \right]} \tag{17}$$

$$= \frac{\exp[z_t^{(1)}]}{\exp[z_t^{(1)}] + \exp[z_t^{(2)}]} \tag{18}$$

where (17) is obtained by applying (14) and (15). We observe that (18) is equivalent to our model $P_{\text{PL}}$ given by (4) for $k = 2$, which proves the proposition. Similarly, we can write $\mathcal{L}_{\text{Indirect}}$ as

$$\mathcal{L}_{\text{Indirect}} = -\sum_t \log P_{\text{PL}}(\pi_t \mid x, y_{<t}) = -\sum_t \log p(y_{\to t}^w \succ y_{\to t}^l|x)$$

where $y_{\to t}^w$ and $y_{\to t}^l$ denote the preferred and dispreferred sub-responses by the teacher.

Note that the preference relation by the teacher is also determined by comparing the logits of the teacher model as follows. Let

$$p_T(y_t|y_{<t}, x) := \frac{\exp(\zeta_t)}{\sum_{\tilde{\zeta}_t} \exp(\tilde{\zeta}_t)}$$

where $\zeta_t$ denotes the *teacher's* logit evaluated at position $t$. Our algorithm determines preference by comparing the teacher's logits. Specifically, let $\zeta_t^{(1)}$ and $\zeta_t^{(2)}$ denote the teacher's logits for subreponses $y_{\to t}^{(1)}$ and $y_{\to t}^{(2)}$ at position $t$. We have that

$$y_{\to t}^{(1)} \succ y_{\to t}^{(2)} \quad \Leftrightarrow \quad \zeta_t^{(1)} \geq \zeta_t^{(2)} \quad \Leftrightarrow \quad r_T(x, y_{\to t}^{(1)}) \geq r_T(x, y_{\to t}^{(2)}) \tag{19}$$

where the if and only if condition on the right can be shown using an argument similar to (14) and (15). This proves the relation between the teacher's reward and its preference in Eq. (8).

In summary, $\mathcal{L}_{\text{Indirect}}$ is equivalent to the negative log-likelihood of preference distribution of sub-responses summed over $t$ under our preference model.

A.2   ANALYSIS OF SOFT TOKEN SELECTION VIA $\sigma_\tau$ IN DIRECT DISTILLATION.

Our claim is based on the gradient rescaling property of KD (Tang et al., 2020). For mathematical tractability, we only analyze forward KL and assume that only the forward KL is used in our direct distillation. Consider the teacher model $p$ and the student model $q$. The student model is an autoregressive LLM whose output is the softmax of token logits. Specifically, let $q_i$ denote the confidence in token $i$ given by

$$q_i = \frac{\exp(z_i)}{\sum_j \exp(z_j)}$$

where $z_i$ denotes the logit of token $i$. Consider the conventional (forward) KD: we minimize the loss

$$L = \mathcal{D}_{KL}(p\|q)$$

If we take the gradient of $L$ with respect to logit $z_i$:

$$\frac{dL}{dz_i} = p_i - q_i \tag{20}$$

This gradient is the "feedback" to logit $z_i$. Now suppose that token $i$ is the "ground truth" for a reasoning task. In reality, there rarely is a ground truth token for LLMs. However, since we consider *reasoning tasks*, there is a highly likely token leading to the correct answer. For example, the next predicted token for `"The answer to 5+3 is  "` is `"8"`. We will assume that *Oracle* exists and provides the one-hot label for the "best token" at each reasoning step. We will simply refer to the best token labeled by Oracle as the ground truth. (20) implies that, if the teacher $p_i$ has a higher confidence than the student $q_i$ ($p_i > q_i$), then the feedback is positive. In that case, training will increase the logit $z_i$ of the ground truth, which is desirable for training. This can speed up the model convergence (Bengio et al., 2009).

Moreover, if the student's confidence is lower (e.g., in the beginning of training), the gradient is relatively stronger. Specifically, consider two students $q$ and $q'$. Suppose $q$ has lower confidence in ground truth token $i$ than $q'$, i.e., $q_i < q'_i$. One can consider $q$ as the student during the early stage of training, and $q'$ as the same model at the later stage. We call $q$ a *weaker* student than $q'$. Consider the ratio of the gradients of $q$ and $q'$ of the logit of $i$-th token:

$$\nabla_i := \frac{dL}{dz_i} \bigg/ \frac{dL'}{dz_i} = \frac{p_i - q_i}{p_i - q'_i} > 1$$

Thus, $\nabla_i > 1$ means that the weaker student or the student in the early training stage achieves an additional multiplicative gain on the gradient. $\nabla_i$ is called the *rescaling factor* of a gradient. The higher rescaling factor can accelerate the training of the student.

Next, we consider the token-selective KD proposed in the paper. We consider "soft selection" based on entropy and sigmoid gating function as in Eq. (10) in the paper. The distillation objective has the following form (only forward KL is used):

$$L_{\text{TS}} = \sigma(H_i(q) - H_i(p)) \cdot D_{KL}(p\|q)$$

where $H_i(p)$ is the entropy of token $i$ evaluated by $p$, and the sigmoid function $\sigma$ enables the "soft-selection" based on the entropy difference. Its gradient is given by

$$\frac{dL_{\text{TS}}}{dz_i} = \sigma(H_i(q) - H_i(p))[p_i - q_i] \tag{21}$$

Note that $\sigma(\cdot)$ term is simply a weight and is computed with no gradient flow to $p$ and $q$ in $\sigma(\cdot)$, i.e., we use "stop-gradient". We can also define the rescaling factor with respect to student models $q$ and $q'$ ($q$ is the weaker student):

$$\nabla_i^{\text{TS}} := \frac{\sigma(H_i(q) - H_i(p))(p_i - q_i)}{\sigma(H_i(q') - H_i(p))(p_i - q'_i)} \tag{22}$$

From (22), we have that

$$\nabla_i^{\text{TS}} = \underbrace{\frac{\sigma(H_i(q) - H_i(p))}{\sigma(H_i(q') - H_i(p))}}_{C} \nabla_i$$

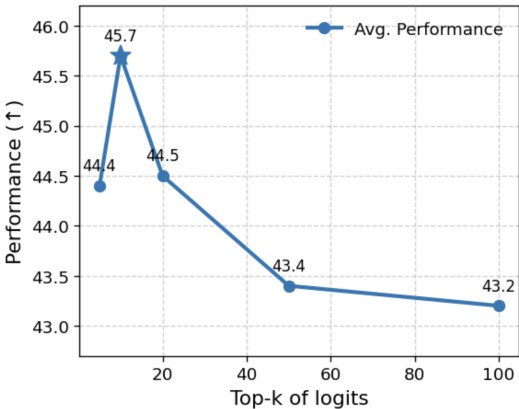

Figure 4: Performances using different top-$k$ of $\mathcal{L}_{\text{Indirect}}$.

Thus, the token selection has a higher rescaling factor than the conventional KD if $C > 1$. We have that

$$C = \frac{\sigma(H_i(q) - H_i(p))}{\sigma(H_i(q') - H_i(p))} = \frac{\exp(-H_i(p)) + \exp(-H_i(q'))}{\exp(-H_i(p)) + \exp(-H_i(q))} \tag{23}$$

If the weaker student $q$ is *less* confident than $q'$ in token $i$, it has *higher* uncertainty. Thus, we $H_i(q) > H_i(q')$, which implies that $C > 1$. Conversely, if the confidence of the student is higher, $C < 1$ from the perspective of that student. Thus, the student will perceive $C$ as being greater than 1 when its confidence is low, e.g., during the early stage of training. Conversely, the student will perceive $C$ as being less than 1 in the later stage when they have higher confidence. This leads to two effects on the student models at different training stages.

**(1) Reinforced gradient.** In the early stage of training, the student model is relatively weak and perceives $C$ as $C > 1$. This means that the scaling factor under token selection is stronger than the conventional KD. Thus, token selection can **reinforce** the gradient towards the "best" token. Such a reinforced gradient helps establish logical abilities in the early stage, facilitating the convergence of training.

**(2) Label smoothing.** In the later stage of the training, the student is well-trained and is likely to have basic reasoning capabilities. The student will perceive $C$ as $C < 1$ compared to the earlier version of itself. This means that the gradient perceived was weaker than the conventional KD. Thus, the student can *diversify* its reasoning process based on already established knowledge. Since the gradient to "best" answer has weakened, we can liken the approach to **label smoothing** (Müller et al., 2019). It allows more exploration of parameter space for reasoning, which is consistent with our student-centric approach.

### A.3 ANALYSIS OF TOP-$k$ IN INDIRECT DISTILLATION.

We investigate the model's performance with varying $k$ in the top-$k$ sub-response selection in indirect distillation. As shown in Fig. 4, we observe that performance peaks at $k = 10$. A smaller number of candidates (e.g., $k = 5$) is insufficient for effective learning. In contrast, performance degrades substantially outside this optimum; for example, increasing $k$ to just 20 causes the score to drop to 44.5.

This result suggests that focused distillation is crucial. Forcing the student to match the teacher's distribution over a wide range of tokens appears to introduce noise in the learning signal. Therefore, we select $k = 10$ as the optimal setting for indirect distillation.

### A.4 ANALYSIS OF THE TOKEN SELECTION RATIO FOR ENTROPY REGULARIZATION

We analyze the effect of the token selection ratio in the entropy regularization. In this section, we will denote the selection ratio by $s\%$. As depicted in Fig. 5, the performance varies significantly with

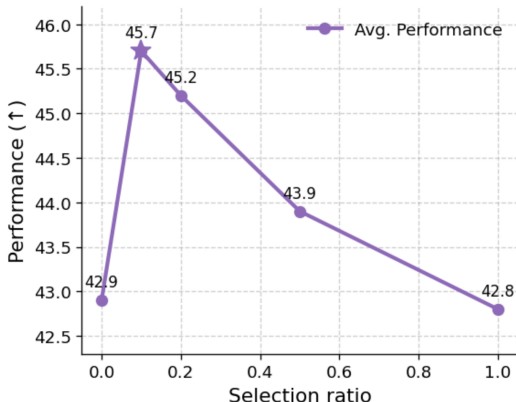

Figure 5: Performances with token selection ratio for entropy regularization.

$s$. With regularization disabled ($s = 0\%$), the model achieves a baseline performance of 42.9. Setting $s = 10\%$ improves the performance to the peak value of 45.7.

However, performance steadily declines for $s$ greater than 20%. At the maximum value of $s = 100\%$, performance drops to 41.8. Our finding indicates that applying entropy minimization across all tokens can be harmful, perhaps because it causes overconfidence of the model.

The results validate our choice of $s = 10\%$ as the optimal setting. This value is also consistent with the observation in (Wang et al., 2025) which estimated the crucial tokens for RL for LLM reasoning to be $10 - 20\%$.

### A.5 ANALYSIS OF HYPERPARAMTER $\beta$.

We vary $\beta$ in JSD($\beta$) from 0.0 to 1.0 and report the average performance on our benchmarks. As illustrated in Fig. 6, the model's performance is highly sensitive to the choice of $\beta$.

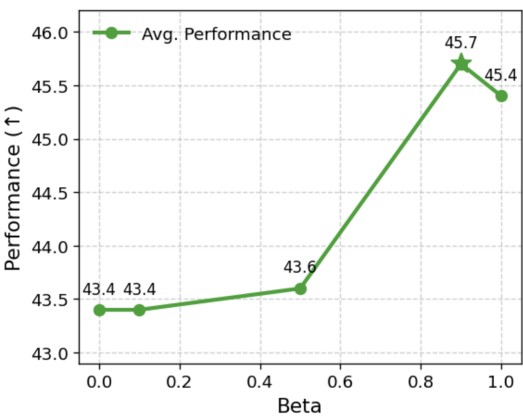

Figure 6: Analysis of $\beta$ of JSD.

When $\beta$ is set to 0, which is equal to forward KL (Hinton et al., 2015), the model achieves a baseline performance of 43.4, and for small values of $\beta$ up to 0.5, we observe only marginal improvements. However, there is a sharp and significant increase in performance as $\beta$ rises from 0.5 to 0.9, where the model achieves its peak performance of 45.7. This substantial gain underscores the critical role of mode-seeking in enhancing the model's reasoning capabilities. Interestingly, increasing $\beta$ further to 1.0 leads to a slight performance degradation to 45.4, suggesting that an excessive weight on this term can disrupt the balance with other training objectives. Based on this empirical evidence, we set $\beta = 0.9$.

| Method | GSM8K | GSM-Plus | MATH | MBPP | IFEval | MMLU-STEM |
|---|---|---|---|---|---|---|
| Off-Policy | 56.0 | 37.1 | 24.9 | 39.8 | 55.0 | 49.5 |
| On-Policy | **60.1** | **40.5** | **26.1** | **40.1** | **55.2** | **50.0** |

Table 6: Performance comparison of on-policy versus off-policy in Qwen2.5 (14B → 1.5B).

| Method | GSM8K | GSM-Plus | MATH | MBPP | IFEval | MMLU-STEM |
|---|---|---|---|---|---|---|
| Qwen2.5-14B (Teacher) | 80.3 | 59.7 | 21.7 | 78.9 | 85.9 | 70.5 |
| Qwen2.5-1.5B (Student) | 57.1 | 38.8 | 16.9 | 38.4 | 53.1 | 49.5 |
| Sequence-Level KD | 56.5 | 38.2 | 16.5 | 40.6 | 53.7 | 48.9 |
| Supervised-KD | 56.3 | 37.8 | 17.9 | 41.0 | 51.4 | 48.4 |
| DistiLLM | 57.2 | 38.7 | 18.6 | 42.1 | 52.5 | 47.8 |
| Speculative KD | 58.0 | 39.2 | 18.5 | 41.6 | 52.4 | 47.8 |
| MiniLLM | 57.7 | 39.7 | 17.8 | **42.2** | 54.7 | 48.2 |
| GKD ($\beta = 0.9$) | 57.9 | 39.9 | 18.1 | 41.8 | 52.3 | 47.7 |
| TSD-KD (Adaptive $c\%$) | 59.3 | 39.8 | **26.5\*** | 39.9 | **55.8** | **50.4** |

Table 7: Performance comparison between distillation methods in Qwen2.5 (14B → 1.5B). Best performance in **Bold** font. (**\***) mark for the student model outperforming the teacher.

| Method | MMLU-Pro-Math | SciQ | BBH | MuSR |
|---|---|---|---|---|
| Qwen2.5-14B (Teacher) | 77.6 | 86.4 | 60.8 | 60.8 |
| Qwen2.5-1.5B (Student) | 34.4 | 85.5 | 36.5 | 38.7 |
| Sequence-Level KD | 32.6 | 87.9 | 36.6 | 36.6 |
| Supervised-KD | 35.3 | 86.6 | 36.4 | 37.9 |
| DistiLLM | 35.2 | 85.7 | 36.5 | 38.7 |
| Speculative KD | 34.8 | 86.7 | 37.2 | 37.7 |
| MiniLLM | 36.2 | 84.1 | 36.0 | 38.3 |
| GKD ($\beta = 0.9$) | 35.7 | 86.6 | 36.2 | 38.5 |
| TSD-KD (Adaptive $c\%$) | **38.5** | **93.8\*** | 40.0 | 39.4 |

Table 8: Performance comparison of more advanced benchmarks in Qwen2.5 (14B → 1.5B). Best performance in **Bold** font. (**\***) mark for the student model outperforming the teacher.

## A.6 ON-POLICY VS. OFF-POLICY DISTILLATION.

To examine the effects of distribution shift, we compare the performance of TSD-KD when trained with on-policy versus off-policy data. The results presented in Table 6 show the superiority of the on-policy framework. The on-policy strategy consistently outperforms its off-policy counterpart across all six benchmarks, achieving an average score of 45.7 versus 43.4, a significant margin of +2.3 points. The difference is particularly high in complex mathematical reasoning tasks such as GSM8K (+4.1) and GSM-Plus (+3.4). These findings suggest that it is more effective to guide the student model along its own generated reasoning paths.

## A.7 ADAPTIVE SELECTION OF $c$.

The adaptive selection of $c$ is performed as follows.

- For each sample $x$, we estimate the sample "difficulty" based on the average uncertainty gap. This is consistent with our direct distillation with the uncertainty gap.

$$C = \sigma(\bar{H}(p_S) - \bar{H}(p_T))$$

where $p_S$, $p_T$ denote the student and teacher model, $\bar{H}$ is the average entropy of each generated sample.

| Method | GSM8K | GSM-Plus | MATH | MBPP | IFEVAL | MMLU-STEM |
|--------|-------|----------|------|------|--------|-----------|
| Supervised-KD | 55.7 | 36.3 | 17.6 | 39.4 | 51.9 | 47.2 |
| DistiLLM | 56.7 | 38.5 | 18.5 | 42.0 | 52.3 | 47.3 |
| MiniLLM | 56.9 | 38.0 | 18.6 | 41.6 | 52.6 | 47.9 |
| GKD ($\beta = 0.9$) | 56.8 | 38.4 | **18.9** | 41.2 | 51.8 | 47.6 |
| TSD-KD | **57.2** | **38.6** | 18.4 | **42.7** | **54.5** | **49.7** |

Table 9: Performance comparison of Parameter Efficient Finetuning between different distillation methods in Qwen2.5.

- The selection ratio $c$ of Opener is determined by the quantile interval of the estimated value of $C$. We take a moving average of $C$ over the window of 5 samples for the estimation of $C$.

$$c = \begin{cases} 5\%, & C \in [0.00, 0.25) \\ 10\%, & C \in [0.25, 0.50) \\ 15\%, & C \in [0.50, 0.75) \\ 20\%, & C \in [0.75, 1.00] \end{cases}$$

Intuitively, a larger uncertainty gap $C$ implies that the current sample is difficult for the student. Thus, we take a larger cumulative percentile (larger $c$) for supervision.

---

**Algorithm 2** Indirect Distillation by Teacher-Guided Preference Ranking

1: **Given**: Teacher model $p_T$, Student Model $p_S^\theta$, Prompt dataset $X$
2: Sample $x$ from $X$, generate $y \sim p_S^\theta(\cdot|x)$
3: c = FIND_C($p_T$, $p_S$)
4: Select the *opener* of $y$ up to top-$c\%$ cumulative entropy
5: **for** each token position $t$ in *opener* **do**
6:    Select top-$k$ logits $\{z_t[1], ..., z_t[k]\}$ by student $p_S^\theta(\cdot|y_{<t}, x)$
7:    Re-rank logits to get the order of preference $\pi_t$ by teacher $p_T(\cdot|y_{<t}, x)$
8:    Compute $P_{\text{PL}}(\pi_t \mid x, y_{<t})$
9: **end for**
10: Compute $\mathcal{L}_{\text{PL}} = -\sum_t \log P_{\text{PL}}(\pi_t | x, y_{<t})$
11: **Function** FIND_C($p_T$, $p_S$)
12:    IF not adaptive
13:        **return** 10%
14:    Calculate an average entropy of generated response, $\bar{H}(p_S)$, $\bar{H}(p_T)$
15:    Compute $C = \sigma(\bar{H}(p_S) - \bar{H}(p_T))$
16:    $c = \begin{cases} 5\%, & C \in [0.00, 0.25) \\ 10\%, & C \in [0.25, 0.50) \\ 15\%, & C \in [0.50, 0.75) \\ 20\%, & C \in [0.75, 1.00] \end{cases}$
17:    **return** $c$
18: **End Function**

---

Overall, TSD-KD with adaptive-$c$ achieves a performance similar to the default setting, i.e., fixed $c$, as shown in Table 7 and 8. Interestingly, the adaptive-$c$ method performs better than the default fixed-$c$ method on MATH and SciQ, which are challenging reasoning benchmarks. Thus, the adaptive-$c$ method is shown to be good at adapting to the difficulty of distillation samples. A pseudocode of Indirect Distillation with adaptive-$c$ is shown in Algorithm 2.

A.8 EFFECTIVENESS IN PARAMETER-EFFICIENT FINETUNING (PEFT).

To assess the practicality and efficiency of our method, we evaluate its performance in a more resource-constrained setting using Parameter-Efficient Finetuning (PEFT). We employ LoRA, a

popular PEFT technique, to update only a small fraction of the student model's parameters during distillation.

The results, summarized in Table 9, demonstrate that TSD-KD's superiority is robustly maintained even in the PEFT paradigm. TSD-KD achieves the highest average score of 43.5, clearly outperforming all competing distillation methods. This strong overall performance is driven by its dominant results across the majority of the benchmarks, securing the top rank in 5 out of 6 tasks, including the challenging reasoning benchmarks GSM8K and MMLU-STEM.

This experiment confirms that the selective learning signal provided by TSD-KD is effective even when updating only a partial set of model parameters. This result highlights the efficiency of our paradigm, which concentrates the distillation process on a crucial subset of tokens. Consequently, TSD-KD can be a more parameter-efficient fine-tuning (PEFT) approach than prior KD methods that operate on the full token set. This efficiency makes TSD-KD a practical and attractive solution for developing capable, small-scale reasoning models under typical computational constraints.

## A.9  HYPERPARAMETERS

| Hyperparameter | Value |
| --- | --- |
| Batch Size | 128 |
| Learning rate | 5e-6 (Qwen2.5), 1e-7 (Gemma2) |
| Learning rate scheduler | Cosine |
| Temperature of KD | 1.0 |
| Max Sequence Length | 1024 |
| Epochs | 3 |
| Optimizer | AdamW |
| Warmup ratio | 0.1 |
| $\alpha$ | 0.1 |
| top-$k$ of $\mathcal{L}_{\text{Indirect}}$ | 10 |
| Selection ratio of $\mathcal{L}_{\text{Indirect}}$ | 0.1 |
| Selection ratio of $\mathcal{L}_{\text{EM}}$ | 0.1 |

Table 10: Detailed hyperparameters.

## A.10  BENCHMARK DETAILS

We provide additional details of the benchmarks used in our evaluation:

GSM8K (Cobbe et al., 2021): A benchmark consisting of grade-school arithmetic word problems, widely used to assess step-by-step mathematical reasoning. Following common practice, we evaluate models in the 5-shot setting as the default.

GSM-Plus (Li et al., 2024): An extension of GSM8K containing more challenging arithmetic word problems, designed to evaluate model robustness beyond simple mathematical reasoning. Following common practice, we adopt the 5-shot setting as the default.

MATH (Hendrycks et al., 2021b): A benchmark of competition-level mathematics problems spanning algebra, geometry, number theory, and combinatorics. Following common practice, we adopt the 4-shot setting as the default.

MMLU-Pro-Math (Wang et al., 2024): A professional-level subset of MMLU-Pro focusing on advanced mathematics questions. Following common practice, we adopt the 5-shot setting as the default.

MMLU-STEM (Hendrycks et al., 2021a): A subset of the Massive Multitask Language Understanding benchmark, covering STEM-related subjects such as physics, chemistry, and biology. Following common practice, we adopt the 5-shot setting as the default.

ScienceQA (SciQ) (Welbl et al., 2017): A science question-answering dataset with multiple-choice format, requiring factual and reasoning skills. Following common practice, we adopt the 0-shot setting as the default.

MBPP (Austin et al., 2021): A program synthesis dataset where models generate short Python functions from natural language problem descriptions. Following common practice, we adopt the 0-shot setting as the default.

BBH (Big-Bench Hard) (Suzgun et al., 2022): A collection of difficult tasks from BIG-Bench that target reasoning and compositional generalization. Following common practice, we adopt the 3-shot setting as the default.

MuSR (Sprague et al., 2024): A benchmark for multi-step soft reasoning in natural language narratives. It is constructed via a neurosymbolic synthetic-to-natural generation process, producing complex scenarios such as long-form murder mysteries. Following common practice, we adopt the 0-shot setting as the default.

IFEval (Zhou et al., 2023): A benchmark that measures instruction-following capabilities, focusing on adherence to task constraints. Following common practice, we adopt the 0-shot setting as the default.

### A.11 MODE COVERING VS. MODE SEEKING

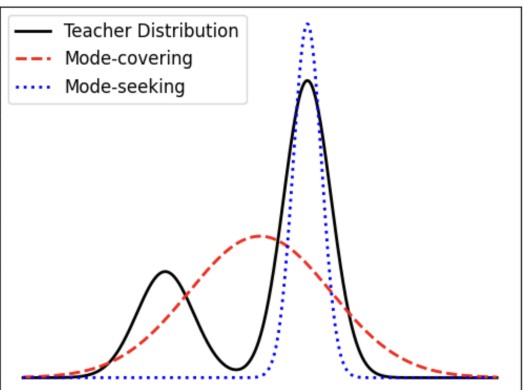

Figure 7: Comparison of mode-covering (forward KL, $\mathcal{D}(P\|Q)$, red) and mode-seeking (reverse KL, $\mathcal{D}(Q\|P)$, blue) behavior. The student ($Q$) is trained to match the bimodal teacher ($P$). The mode-seeking distribution fits one mode precisely, while the mode-covering distribution spreads its mass to cover both. Adapted from (Le, 2017).

Figure 7 illustrates the difference between mode-covering (forward KL) and mode-seeking (reverse KL) when distilling a multi-modal teacher distribution ($P$) into a unimodal student distribution ($Q$) with limited capacity. These observations are based on and are also available in the previous work (Le, 2017; Agarwal et al., 2024; Gu et al., 2023).

- Mode-seeking (Reverse KL, $\mathcal{D}(Q\|P)$) heavily penalizes the student ($Q$) for placing probability mass where the teacher ($P$) is zero. This forces the student to focus on fitting high-probability modes of the teacher's distribution.

- Mode-covering (Forward KL, $\mathcal{D}(P\|Q)$), in contrast, penalizes the student for not assigning the probability mass where the teacher's distribution is nonzero. This encourages the student to generate distribution that broadly covers the teacher's modes.

### A.12 QUALITATIVE ANALYSIS

We present **qualitative examples** in two distinct domains: mathematical reasoning and code generation (see Fig. 8 and Fig. 9). We make comparisons with strong baselines (Qwen2.5-1.5B) and competing methods (GKD).

**Case 1: Task Alignment in Code Generation.** In the second example (Fig. 8), the user explicitly requests a function that converts a string to lower case.

- **Baselines:** Both baselines incorrectly interpret the task as verification rather than transformation, generating code that checks the condition (e.g., `s.islower()`) instead of performing the requested transformation. This results in semantically incorrect outputs.

- **Ours:** Despite the ambiguous context, which can often trigger a check-function pattern, our model accurately identifies the user's intent to manipulate the input string. It produces the correct transformation logic (e.g., `s.lower()`), demonstrating superior alignment with user intent beyond superficial pattern matching.

**Case 2: Constraint Satisfaction in Mathematical Reasoning.** In the first example (Fig. 9), the prompt requires a multi-step calculation followed by a specific output-format constraint ("in cents").

- **Baselines:** Qwen2.5-1.5B correctly computes the arithmetic ($3 - 2.4 = 0.6$) but fails the unit-conversion constraint, outputting "0.6 dollars" instead of converting to cents. GKD, in contrast, exhibits generation collapse and fails to complete the chain of thought.
- **Ours:** Our model maintains contextual constraints throughout the reasoning process, successfully performing the arithmetic and adhering to the formatting requirement, ultimately producing "60 cents."

---

**Q:** Write a python function to convert the given string to lower case.

---

**Qwen2.5-1.5B:**

```
import re

def is_lower(s):
    # Check if all characters in the string are lowercase
    return s.islower()
```

**GKD:**

```
import re

def is_lower(s):
    # Check if all characters in the string are lowercase
    return s.islower()
```

**Ours:**

```
import re

def is_lower(s):
    s = str.lower(s)
    return s
```

---

Figure 8: Qualitative Analysis of Code generation (MBPP).

**Q:** Trey receives a 5% commission on every sale he makes. On the sale of a \$60 coat (before any discounts), how many more cents will he receive if his commission is based on the original price of the coat rather than the price of the coat after a 20% discount?

---

**GT:** 60

---

**Qwen2.5-1.5B:**

To determine how much more Trey will receive as a commission if the sale is based on the original price of the coat rather than the discounted price, we need to calculate the commissions for both scenarios and then find the difference.

First, let's calculate the commission based on the original price of the coat:
The original price of the coat is \$60. With a 5% commission rate, the commission Trey would receive is:

$$60 \times 0.05 = 3 \text{ dollars}$$

Next, let's calculate the commission based on the discounted price of the coat:
The coat is sold at a 20% discount, so the selling price before tax is:

$$60 \times (1 - 0.20) = 60 \times 0.80 = 48 \text{ dollars}$$

With a 5% commission rate, the commission Trey would receive is:

$$48 \times 0.05 = 2.4 \text{ dollars}$$

Now, we find the difference between the two commissions:

$$\textcolor{red}{3 - 2.4 = 0.6 \text{ dollars}}$$

Since the question asks for the difference

---

**GKD:**

To determine how much more Trey will receive as a commission when the sale is based on the original price of the coat rather than the discounted price, let's follow these steps:
1. Calculate the original price of the coat before any discount:
The original price of the coat is \$60.
2. Determine the commission based on the original price:
Trey receives a 5% commission on the original price of the coat. So, the commission from the original price is calculated as follows:
Commission from original price = 0.05 x 60 = 3 dollars
3. Calculate the price of the coat after a 20% discount:
A 20% discount means the customer pays 80% of the original price. Thus, the discounted price is: Discounted price = 60 x (1 - 0.20) = 60 x 0.80 = 48 dollars
4. Determine the commission based on the discounted price: With the discounted price, the commission is
Not complete

---

**Ours:**

The original price of the coat is 60 dollars. Trey gets a 5% commission on sales. So he gets $60 \times .05 = 3$ dollars from the sale.
Now if the price was discounted by 20% , the new price would be $60(1 - .2) = 48$ dollars. He would get $48 \times .05 = 2.4$ dollars from the sale.
So he gets $3 - 2.4 = 0.6$ dollars more when the price is not discounted.
In cents, this is $0.6 \times 100 = 60$ cents.
Therefore, he will receive 60 more cents.
Final Answer: The final answer is 60 cents.

Figure 9: Qualitative Analysis of Math solution (MATH).

