# OpenReview forum: "Explain in Your Own Words: Improving Reasoning via Token-Selective Dual Knowledge Distillation"
_ICLR.cc/2026/Conference — ICLR 2026 Poster_

### Official Review · Reviewer_DKho · 2025-10-25

**Soundness:** 2
**Presentation:** 3
**Contribution:** 2
**Rating:** 4
**Confidence:** 4

**Summary:**

This paper proposes Token-Selective Dual Knowledge Distillation (TSD-KD), a student-centric framework for improving reasoning through selective and dual-mode knowledge transfer. The method integrates: Indirect distillation, the student proposes candidate tokens and receives preference ranking feedback from the teacher (similar to DPO-like weak supervision); Direct distillation, selective distribution matching on tokens with large student–teacher uncertainty gaps using JSD, Entropy regularization — confidence enhancement through minimizing entropy of the most uncertain tokens. The authors conduct comprehensive experiments on reasoning benchmarksusing Qwen2.5 and Gemma2 model families. Results show consistent improvements, with the student model occasionally outperforming its teacher.

**Strengths:**

1. Complete and sound framework combining distillation with entropy regularization forms a coherent pipeline.

2. Writing is good.

3. Use of token entropy for identifying important tokens aligns with recent research trends in reasoning-focused LLMs (e.g., 80/20 entropy rule, ARPO).

4. Comprehensive experiments across two model families demonstrate generalizability.

**Weaknesses:**

1. The core idea of TSD-KD is highly similar to <Keypoint-based Progressive Chain-of-Thought Distillation (icml 2024)> in motivation. Both approaches emphasize selective token weighting and distillation; TSD-KD replaces KPOD’s mask-learning with entropy-based selection but retains the same underlying philosophy. However, this previous work is completely neglected.


2. At this time, it is unclear why the authors did not conduct experiments on the Qwen3 family, such as Qwen3-8B, which has become the de-facto principle for reasoning evaluation (like in <Beyond the 80/20 Rule: High-Entropy Minority Tokens Drive Effective Reinforcement Learning for LLM Reasoning>.). The paper only reports results on Qwen2.5 and Gemma2, both of which are now relatively outdated and substantially weaker in reasoning capability. Since the proposed method explicitly targets reasoning enhancement, it is essential to verify its effectiveness on more competitive and up-to-date models.

3. Limited theoretical insight are provided in this paper. For example, the paper lacks a clear theoretical justification for why entropy-based token selection truly improves reasoning robustness beyond serving as a heuristic importance measure.

**Questions:**

refer to Weaknesses, I will adjust my score according to the responses.

---

> ### Author Response · Authors · 2025-11-24
> **Response to Reviewer DKho (1/3)**
>
> We sincerely appreciate the reviewer for valuable comments. We have addressed all of the concerns raised by the reviewer.
>
> ---
>
> ***W1: The core idea of TSD-KD is highly similar to <Keypoint-based Progressive Chain-of-Thought Distillation (icml 2024)>. in motivation. Both approaches emphasize selective token weighting and distillation; TSD-KD replaces KPOD’s mask-learning with entropy-based selection but retains the same underlying philosophy. However, this previous work is completely neglected.***
>
> Thank you for pointing this work out. We will discuss KPOD in Related Work in the revised paper as follows.
>
> ***KPOD [1] proposed a progressive off-policy KD that transfers the teacher’s CoT reasoning to the student by selective distillation of key tokens combined with curriculum learning. In contrast, we take an on-policy, dual distillation approach with token selection based on entropy.***
>
> Below we summarize the key differences between KPOD and our method.
>
> **1. Problem settings: off-policy rationale imitation vs. on-policy preference-guided refinement.**
> - KPOD is an off-policy, teacher-centric rationale imitation method: the student is trained to reproduce teacher-generated chain-of-thought traces.
>
> - TSD-KD is on-policy and student-centric: the student generates its own reasoning, and the teacher provides only weak preference signals rather than rationales.
>
> 2. **Token-selection criteria.**
> - KPOD identifies keypoints via a learned mask over the teacher’s rationale, optimized for answer reconstruction.
> - TSD-KD selects tokens based on *student-side uncertainty*, including (a) opener tokens detected by cumulative entropy and (b) tokens with large teacher–student entropy gaps, without relying on the teacher’s rationale structure.
>
> 3. **Absence of indirect distillation in KPOD**
> - TSD-KD incorporates preference-based indirect supervision using the Plackett–Luce ranking model [2], where the teacher re-ranks the student’s candidate tokens.
> - KPOD does not use ranking-based or indirect supervision.
>
> 4. **Distillation philosophy**
> - KPOD: reproduction of the teacher’s rationale.
> - TSD-KD: reasoning improvement with minimal teacher intervention.
>
> **Comparison.** The official KPOD code has not been released; therefore, we omit experimental comparisons.
>
> **References**
>
> [1] Feng, Kaituo, et al. Keypoint-based progressive chain-of-thought distillation for LLMs. arXiv preprint arXiv:2405.16064 (2024).
>
> [2] Plackett, Robin L. The analysis of permutations. Journal of the Royal Statistical Society Series C: Applied Statistics 24.2 (1975): 193–202.
>
> ---
>
> ***W2: At this time, it is unclear why the authors did not conduct experiments on the Qwen3 family, such as Qwen3-8B, which has become the de-facto principle for reasoning evaluation (like in <Beyond the 80/20 Rule: High-Entropy Minority Tokens Drive Effective Reinforcement Learning for LLM Reasoning>.). The paper only reports results on Qwen2.5 and Gemma2, both of which are now relatively outdated and substantially weaker in reasoning capability. Since the proposed method explicitly targets reasoning enhancement, it is essential to verify its effectiveness on more competitive and up-to-date models.***
>
>
> Thank you for the suggestion. We conducted experiments on Qwen3 (8B as a teacher and 1.7B as a student) to verify the effectiveness of our method on more competitive and up-to-date reasoning models. The results are summarized in Table 1.
>
> | Method                 | GSM8K | MATH | GSM_Plus | MBPP | IFEVAL | MMLU_STEM |
> |------------------------|-------|------|----------|------|--------|-----------|
> | **Qwen3-8B (teacher)** | 82.9  | 22.4 | 59.4     | 66.0 | 86.5   | 40.2      |
> | **Qwen3-1.7B (student)** | 66.3 | 21.0 | 43.1     | 50.8 | 75.1   | 36.2      |
> | Sequence KD            | 65.7  | 20.8 | 43.5     | 51.3 | 75.7   | 36.9      |
> | Supervised KD          | 66.5  | 21.5 | 43.7     | 51.2 | 76.1   | 36.5      |
> | DistiLLM               | 66.7  | 21.2 | 45.5     | 51.4 | 78.2   | 36.8      |
> | MiniLLM                | 67.0  | 22.6 | 46.4     | **52.1** | 79.2 | 36.5      |
> | GKD                    | 67.2  | 22.8 | 46.0     | 51.7 | 79.1   | 36.7      |
> | **TSD-KD**             | **68.7** | **28.0** | **47.2** | 51.9 | **79.8** | **37.5** |
>
> Table 1.
>
> We observe a similar trend in performance as Qwen2.5 (Table 1 in the paper). Our method achieves state-of-the-art performance in most of the benchmarks. Moreover, on the challenging **MATH** benchmark, the student model from our method **outperforms the teacher model by a large margin (+25.0\%)**; this improvement is even greater than the case of Qwen 2.5 **(+20.3\%)** on **MATH**. These results confirm that our method consistently improves reasoning performance across multiple benchmarks, demonstrating its effectiveness on more competitive reasoning models like Qwen3-8B.
>
> **Action:** The experimental results have been added to the revised paper. The discussion is at line 466, and the results are in Table 5.

---

> ### Author Response · Authors · 2025-11-24
> **Response to Reviewer DKho (2/3)**
>
> ***W3: Limited theoretical insight are provided in this paper. For example, the paper lacks a clear theoretical justification for why entropy-based token selection truly improves reasoning robustness beyond serving as a heuristic importance measure.***
>
> We provide a theoretical analysis of the benefits of token selection on training stability.
>
> **Claim: Token selection can scale the gradient of the student model during training, facilitating convergence.**
>
> Our claim is based on the gradient rescaling property of KD [1]. For mathematical tractability, we only analyze forward KL, and assume that only the forward KL is used in our direct distillation. Consider the teacher model $p$ and the student model $q$. The student model is an autoregressive LLM whose output is the softmax of token logits. Specifically, let $q_i$ denote the confidence in token $i$ given by
>
> $$
> q_i = \frac{\exp(z_i)}{\sum_j \exp(z_j)}
> $$
>
> where $z_i$ denotes the logit of token $i$. Consider the conventional (forward) KD: we minimize the loss
>
> $$
> L = \mathcal D_{KL}(p\|q)
> $$
>
> If we take the gradient of $L$ with respect to logit $z_i$:
>
> $$
> \frac{dL}{dz_i} = p_i - q_i \tag{1}
> $$
>
> This gradient is the *feedback* to logit $z_i$. Now suppose that token $i$ is the *ground truth* for a reasoning task. In reality, there rarely is a ground truth token for LLMs. However, since we consider *reasoning tasks*, there is a highly likely token leading to the correct answer. For example, the next predicted token for **The answer to 5+3 is** is **8**. We will assume that *Oracle* exists and provides the one-hot label for the "best token" at each reasoning step. We will simply refer to the best token labeled by Oracle as the ground truth. Equation (1) implies that, if the teacher $p_i$ has a higher confidence than the student $q_i$, $(p_i>q_i)$, then the feedback is positive. In that case, training will increase logit $z_i$ of the ground truth, which is desirable for training. This can speed up the model convergence [2].
>
> Moreover, if the student's confidence is lower (e.g., in the beginning of training), the gradient is relatively stronger. Specifically, consider two students $q$ and $q'$. Suppose $q$ has lower confidence on ground truth token $i$ than $q'$, i.e., $q_i < q_i'$. One can consider $q$ as the student during the early stage of training, and $q'$ as the same model at the later stage. We call $q$ a *weaker* student than $q'$. Consider the ratio of the gradients of $q$ and $q'$ of the logit of the $i$-th token:
>
> $$
> \nabla_i := \frac{dL/dz_i}{dL'/dz_i} = \frac{p_i - q_i}{p_i - q'_i} > 1
> $$
>
> Thus, $\nabla_i>1$ means that the weaker student or the student in the early training stage achieves an additional multiplicative gain on the gradient. $\nabla_i$ is called the *rescaling factor* of a gradient. The higher rescaling factor can accelerate the training of the student.
>
> Next, we consider the token-selective KD proposed in the paper. We consider "soft selection" based on entropy and sigmoid gating function as in Eq. (10) in the paper. The distillation objective has the following form (only forward KL is used):
>
> $$
> L_{\text{TS}} = \sigma(H_i(q)-H_i(p)) \cdot \mathcal D_{KL}(p\|q)
> $$
>
> where $H_i(p)$ is the entropy of token $i$ evaluated by $p$, and the sigmoid function $\sigma$ enables the "soft selection" based on the entropy difference. Its gradient is given by
>
> $$
> \frac{dL_{\text{TS}}}{dz_i}
> = \sigma(H_i(q)-H_i(p))[p_i - q_i] \tag{2}
> $$
>
> Note that the $\sigma(\cdot)$ term is simply a weight and is computed with no gradient flow to $p$ and $q$ in $\sigma(\cdot)$ (we apply stop-gradient). We can also define the rescaling factor with respect to student models $q$ and $q'$ ($q$ is the weaker student):
>
> $$
> \nabla_i^\text{TS} :=
> \frac{\sigma(H_i(q)-H_i(p))(p_i-q_i)}
> {\sigma(H_i(q')-H_i(p))(p_i-q_i')} \tag{3}
> $$
>
> From (3), we have that
>
> $$
> \nabla_i^{\textrm{TS}}=\underbrace{
> \frac{\sigma\left(H_i(q)-H_i(p)\right)}
> {\sigma\left(H_i(q')-H_i(p)\right)}
> }_{C}
> \nabla_i
> $$
>
>
> Thus, the token selection has a higher rescaling factor than the conventional KD if $C>1$. We have that
>
> $$
> C =\frac{\sigma(H_i(q)-H_i(p))}
> {\sigma(H_i(q')-H_i(p))}
> =\frac{\exp(-H_i(p)) + \exp(-H_i(q'))}
> {\exp(-H_i(p)) + \exp(-H_i(q))} \tag{4}
> $$
>
> If the weaker student $q$ is *less* confident than $q'$ in token $i$, it has *higher* uncertainty. Thus, $H_i(q) > H_i(q')$, which implies that $C>1$. Conversely, if the confidence of the student is higher, $C<1$. Thus, the student will perceive $C>1$ during the early stage of training, while it will perceive $C<1$ during the later stage. This leads to two effects on the student models at different stages of training.
>
> (continue)

---

> ### Author Response · Authors · 2025-11-24
> **Response to Reviewer DKho (3/3)**
>
> **(1) Reinforced gradient.** In the early stage of training, the student model is relatively weak and perceives $C$ as $C>1$. This means that the scaling factor under token selection is stronger than the conventional KD. Thus, token selection can **reinforce** the gradient towards the "best" token. Such a reinforced gradient helps establish logical abilities in the early stage, facilitating the convergence of training.
>
> **(2) Label smoothing.** In the later stage of the training, the student is well-trained and is likely to have basic reasoning capabilities. The student will perceive $C$ as $C<1$ compared to the earlier version of itself. This means that the gradient is perceived weaker than the conventional KD. Thus, the student can *diversify* its reasoning process based on already established knowledge. Since the gradient to "best" answer has weakened, we can liken the approach to **label smoothing** [3]. It allows more exploration of parameter space for reasoning, which is consistent with our student-centric approach.
>
> **Action:** We will add the analysis to Appendix A.8 (due to space limitation). Please see Appendix A.8 of the revised paper.
>
> **References**
>
> [1] Tang, Jiaxi, et al. "Understanding and improving knowledge distillation." arXiv preprint arXiv:2002.03532 (2020).
>
> [2] Yoshua Bengio, Jérôme Louradour, Ronan Collobert, and Jason Weston. Curriculum learning. In
> Proceedings of the 26th annual international conference on machine learning, pp. 41–48. ACM,
> 2009.
>
> [3] Müller, Rafael, Simon Kornblith, and Geoffrey E. Hinton. "When does label smoothing help?." Advances in neural information processing systems 32 (2019).

---

> > ### Comment · Reviewer_DKho · 2025-11-25
> >
> > Thanks for addressing my concerns. I have increased my score accordingly.

---

> ### Author Response · Authors · 2025-11-25
> **Thank you for raising the score (4->6)**
>
> Dear Reviewer,
>
> Thank you so much for acknowledging our responses and raising the score.
>
> Your comments have been immensely helpful for improving our work.
>
> Authors

---

### Official Review · Reviewer_9rZk · 2025-10-27

**Soundness:** 3
**Presentation:** 3
**Contribution:** 3
**Rating:** 6
**Confidence:** 3

**Summary:**

This paper introduces Token-Selective Dual Knowledge Distillation (TSD-KD), a framework for distilling reasoning ability from large language models to smaller student models by focusing on important tokens in the reasoning chain. TSD-KD combines two main innovations: (1) indirect, preference-based distillation, where the teacher re-ranks student-generated token candidates without forcing its output distribution, and (2) direct, gated distillation that selectively applies distribution-matching to tokens where the student is uncertain and the teacher is confident. The approach is regularized with selective entropy minimization on the most uncertain tokens. Empirical evaluation across 10 challenging reasoning benchmarks demonstrates TSD-KD’s strong performance, including cases where the compressed student surpasses its teacher.

**Strengths:**

1.The use of student-generated candidates (preference-based indirect distillation) and selective, entropy-based token gating in direct distillation is thoughtfully motivated and distinguishes the framework from prior “teacher-forcing” approaches. The focus on letting the student “explain in its own words” resonates with cognitive insights and supports the central claim.
2.The explicit combination of indirect and direct knowledge distillation, each carefully limited to critical tokens, is well-positioned to address known weaknesses of pure distribution-matching or of-point-wise imitation.
3. The mathematical formulation is transparent, the underlying assumptions are stated, and the algorithmic components are described with appropriate rigor.
4. TSD-KD consistently outperforms strong baselines, with substantial absolute gains. Importantly, in multiple cases, the student model trained via TSD-KD surpasses the teacher.

**Weaknesses:**

1. The preference-based indirect distillation encourages the student to align with the teacher’s ranking on top-$k$ student candidates. However, this assumes that the student's beam search is likely to generate candidates close to the correct reasoning trace, which may not hold for weaker students or for highly ambiguous problems.
2. While tables and figures provide extensive quantitative results, the paper lacks qualitative or error analysis on the types of reasoning improvements the student makes with TSD-KD (beyond aggregate accuracy).
3. Even though performance improvements are noticeable, the paper does not report any statistical significance tests.

**Questions:**

1. How robust is the preference-based indirect distillation if the student’s top-$k$ candidates are mostly incorrect? Does the framework degrade gracefully if initial reasoning is off-policy, or does performance collapse? Are there analyses on very weak students or pathological candidate proposals?
2. Do any ablation results suggest redundancy between direct distillation with entropy gating and selective entropy minimization? Are there tasks where one suffices without the other, and can the gains be attributed to only one component in certain domains?
3. What steps are in place to detect or mitigate biases propagated from teacher to student, considering that only a subset of tokens is distilled but on the student’s own output?

---

> ### Author Response · Authors · 2025-11-27
> **Response to Reviewer 9rZk (1/4)**
>
> We sincerely appreciate the reviewer for valuable comments. We have addressed all of the concerns raised by the reviewer.
>
> ---
>
> ***W1: The preference-based indirect distillation encourages the student to align with the teacher’s ranking on top-k
>  student candidates. However, this assumes that the student's beam search is likely to generate candidates close to the correct reasoning trace, which may not hold for weaker students or for highly ambiguous problems.***
>
> ***Q1: How robust is the preference-based indirect distillation if the student’s top-k candidates are mostly incorrect? Does the framework degrade gracefully if initial reasoning is off-policy, or does performance collapse? Are there analyses on very weak students or pathological candidate proposals?***
>
> *Initially, we prepared the responses to **W1** and **Q1** separately. We found that the responses substantially overlapped. Thus, we will provide the combined response to **W1** and **Q1** as follows.*
>
> Following your suggestion, we evaluate our method with **Qwen2.5-0.5B**, a substantially weaker student model than our baseline LLMs (e.g., 1.5B$\sim$2B). The model is more likely to produce proposals that diverge from the teacher’s top-$k$ tokens.
> Table 1 shows the results. We compared Supervised KD (off-policy), GKD (on-policy, direct) and TSD-KD (on-policy, indirect + direct).
>
> | Method                          | GSM8K | MATH | IFEVAL | MMLU STEM |
> |---------------------------------|-------|------|--------|-----------|
> | Qwen2.5-0.5B                    | 33.1  | 10.2 | 39.1   | 35.0      |
> | Supervised KD (Off-policy)      | **34.4** | 11.1 | 39.9   | 33.8      |
> | GKD (On-policy, Direct)         | 34.1  | 11.5 | 39.3   | 35.3      |
> | TSD-KD (On-policy, Indirect + Direct) | 34.2  | **13.4** | **41.2** | **35.7** |
>
> Table 1.
>
> **1. Direct distillation also struggles with weak student.** The results show that off-policy (supervised KD) or On-policy with direct distillation only (GKD) did not perform well compared to our method. On GSM8K, Supervised-KD performed the best but only 0.2\% difference from our method. Our method consistently outperformed others over the rest of the benchmarks. The gain is the greatest on **MATH** benchmark which is particularly challenging for a 0.5B model. When the student is weaker, the **distribution shift** from the strong distillation will even worse, because the capacity gap between the teacher and student is greater. This is particularly true for challenging reasoning tasks like **MATH** benchmark.
>
>
> **2. Dual (Indirect+Direct) Distillation is robust and effective for weak students.** Since our method uses dual (indirect+ direct) distillation, the superiority over direct distillation indicates that indirect distillation actually helps improve reasoning. The following are possible explanations.
>
> - Firstly, indirect distillation can mitigate the distribution shift problem by performing a weak form of distillation. Thus, it complements direct distillation, which is a strong form of distillation.
>
> - Secondly, although the student's top-$k$ may not match those of the teacher, the teacher's fine-grained ranking in indirect distillation may reinforce "better" reasoning trajectories while suppressing the most severe hallucinations. This can be a proper signal for the student to evolve into a better reasoner. The gains on **MATH** or **IFEVAL** support the findings.
>
> - Thirdly, indirect distillation is complemented by direct distillation. Direct distillation can cover "difficult tokens" that may not have been properly supervised by Indirect Distillation. This is the main motivation of our dual distillation: **Indirect and direct distillation work in a complementary manner**. This dual approach is key to maintaining consistency and quality of reasoning.
>
> **Conclusion.**
>
> Consistent improvements across diverse benchmarks with the small 0.5B student confirm that the performance of TSD-KD degrades gracefully with the model capacity and avoids the performance collapse.

---

> ### Author Response · Authors · 2025-11-27
> **Response to Reviewer 9rZk (2/4)**
>
> ***W2: While tables and figures provide extensive quantitative results, the paper lacks qualitative or error analysis on the types of reasoning improvements the student makes with TSD-KD (beyond aggregate accuracy).***
>
> ## Case 1. (Math)
>
> | Section | Content |
> |--------|---------|
> | **Q** | **Trey receives a 5\% commission on every sale he makes. On the sale of a \$60 coat (before any discounts), how many more cents will he receive if his commission is based on the original price of the coat rather than the price of the coat after a 20\% discount?** |
> | **GT** | 60 |
> | **Qwen2.5-1.5B** | See block below. |
> | **GKD** | See block below. |
> | **Ours** | See block below. |
>
> ---
>
> ### Qwen2.5-1.5B (Conversion Error)
> ```python
> To determine how much more Trey will receive as a commission if the sale is based on the original price of the coat rather than the discounted price, we need to calculate the commissions for both scenarios and then find the difference.
>
> First, let's calculate the commission based on the original price of the coat:
>
> The original price of the coat is \$60. With a 5\% commission rate, the commission Trey would receive is:
> 60 x 0.05 = 3 dollars
>
> Next, let's calculate the commission based on the discounted price of the coat:
>
> The coat is sold at a 20\% discount, so the selling price before tax is:
> 60 x( 1 - 0.20) = 60 \times 0.80 = 48 dollars
> With a 5\% commission rate, the commission Trey would receive is:
> 48 x 0.05 = 2.4 dollars
>
> Now, we find the difference between the two commissions:
> 3 - 2.4 = 0.6 dollars
> ```
>
> ### GKD (Incorrect)
> ```python
> To determine how much more Trey will receive as a commission when the sale is based on the original price of the coat rather than the discounted price, let's follow these steps:
>
> 1. Calculate the original price of the coat before any discount:
>
>    The original price of the coat is \$60.
>
> 2. Determine the commission based on the original price:
>
> Trey receives a 5\% commission on the original price of the coat. So, the commission from the original price is calculated as follows:
>
> Commission from original price = 0.05 x 60 = 3 dollars
>
> 3. Calculate the price of the coat after a 20\% discount:
>
> A 20\% discount means the customer pays 80\% of the original price. Thus, the discounted price is:
> Discounted price = 60 x (1 - 0.20) = 60 x 0.80 = 48 dollars
> ```
>
> ### Ours (Correct)
> ```python
> The original price of the coat is $60$ dollars. Trey gets a $5\%$ commission on sales. So he gets $60\times .05=3$ dollars from the sale.
>
> Now if the price was discounted by 20\% , the new price would be 60(1-.2)=48 dollars. He would get 48 x .05=2.4 dollars from the sale.
>
> So he gets 3-2.4=0.6 dollars more when the price is not discounted.
>
> In cents, this is 0.6\times 100=60 cents.
>
> Therefore, he will receive $60$ more cents.
>
> Final Answer: The final answer is 60 cents.
> ```
>
> ---
>
> ## Case 2. (Code)
>
> | Section | Content |
> |--------|---------|
> | **Q** | **Write a python function to convert the given string to lower case.** |
> | **Qwen2.5-1.5B** | See code block below. |
> | **GKD** | See code block below. |
> | **Ours** | See code block below. |
>
> ---
>
> ### Qwen2.5-1.5B (Fail)
> ```python
> import re
>
> def is_lower(s):
>     # Check if all characters in the string are lowercase
>     return s.islower()
> ```
>
> ### GKD (Fail)
> ```python
> import re
>
> def is_lower(s):
>     # Check if all characters in the string are lowercase
>     return s.islower()
> ```
>
> ### Ours (Pass)
> ```python
> import re
>
> def is_lower(s):
>     s = str.lower(s)
>     return s
> ```
>
> (continue)

---

> ### Author Response · Authors · 2025-11-27
> **Response to Reviewer 9rZk (3/4)**
>
> We present **qualitative examples** in two distinct domains: mathematical reasoning and code generation (see above cases). We make comparisons with strong baselines (Qwen2.5-1.5B) and competing methods (GKD).
>
> **Case 1: Constraint Satisfaction in Mathematical Reasoning.**
> In the first example (Math case), the prompt requires a multi-step calculation followed by a specific output-format constraint ("in cents").
>
> **Baselines:** Qwen2.5-1.5B correctly computes the arithmetic ($3-2.4=0.6$) but fails the unit-conversion constraint, outputting "0.6 dollars" instead of converting to cents. GKD, in contrast, exhibits generation collapse and fails to complete the chain of thought.
> - **Ours:** Our model maintains contextual constraints throughout the reasoning process, successfully performing the arithmetic and adhering to the formatting requirement, ultimately producing "60 cents."
>
>
> **Case 2: Task Alignment in Code Generation.**
> In the second example (Code), the user explicitly requests a function that converts a string to lower case.
>
> - **Baselines:** Both baselines incorrectly interpret the task as verification rather than transformation, generating code that checks the condition (e.g., **s.islower()**) instead of performing the requested transformation. This results in semantically incorrect outputs.
>  - **Ours:** Despite the ambiguous context, which can often trigger a check-function pattern, our model accurately identifies the user's intent to manipulate the input string. It produces the correct transformation logic (e.g., **s.lower()**), demonstrating superior alignment with user intent beyond superficial pattern matching.
>
> **Conclusion.**
> The results demonstrate that while baselines may excel in surface-level fluency, they struggle with precise constraint satisfaction (Case 1) and semantic intent disambiguation (Case 2). In contrast, our method mitigates these issues, producing more reliable and actionable outputs across diverse tasks.
>
> **Action:** We added the qualitative results to the revised paper in Appendix A.9.
>
> ---
>
> ***W3: Even though performance improvements are noticeable, the paper does not report any statistical significance tests.***
>
> **1. Absence of statistical tests.** We explain why there were no statistical tests. The default pipeline for evaluation is deterministic by design. Specifically, all evaluation results are obtained using greedy decoding (temperature = 0). Therefore, given a fixed model checkpoint and prompt, the generation outputs and resulting accuracies are identical across runs. Thus, traditional significance tests (e.g., t-test over random seeds) are not directly applicable or meaningful. Similar deterministic settings for evaluation are used in recent KD/LLM works [1] [2] [3].
>
> Instead, we evaluated cross-task and cross-model consistency of the model's performance. The same TSD-KD setting achieves consistent improvements across 10 diverse reasoning benchmarks and three model architectures (**Qwen2.5**, **Qwen3.0** and **Gemma2**), demonstrating the robustness of the observed gains.
>
> **2. Bootstrap Resampling Test.** We can use bootstrap resampling for a statistical significance test of the performance differences between our method and baselines. Table 2 shows the $p$-value of bootstrap resampling test, with sample size 100 and repetition number=10,000. With significance level of 0.05, we can conclude that the superiority of our method over the baselines is statistically significant $(p<0.05)$ across diverse benchmarks.
>
> | method KD            | MATH   | MBPP   | IFEVAL |
> | -------------------- | ------ | ------ | ------ |
> | Ours > Supervised KD | 0.0006 | 0.0002 | 0.003  |
> | Ours > MiniLLM       | 0.0151 | 0.0003 | 0.005  |
> | Ours > GKD           | 0.0134 | 0.0001 | 0.001  |
>
> **Table 2** p-value of Bootstrap Resampling (sample size=100), when repetition number=10,000
>
> **Action.** We added the statistical significance test in Appendix A.9.
>
> **References**
>
> [1] Agarwal, Rishabh, et al. "On-policy distillation of language models: Learning from self-generated mistakes." The twelfth international conference on learning representations. 2024.
>
> [2] Yang, An, et al. "Qwen3 technical report." arXiv preprint arXiv:2505.09388 (2025).
>
> [3] Comanici, Gheorghe, et al. "Gemini 2.5: Pushing the frontier with advanced reasoning, multimodality, long context, and next generation agentic capabilities." arXiv preprint arXiv:2507.06261 (2025).

---

> ### Author Response · Authors · 2025-11-27
> **Response to Reviewer 9rZk (4/4)**
>
> ***Q2: Do any ablation results suggest redundancy between direct distillation with entropy gating and selective entropy minimization? Are there tasks where one suffices without the other, and can the gains be attributed to only one component in certain domains?***
>
> | Entropy gating | $\mathcal{L}_\text{EM}$ | GSM8K | MATH | IFval | MMLU-STEM |
> | -------------- | ----------------------- | ----- | ---- | ----- | --------- |
> | ✗              | ✗                       | 57.9  | 18.1 | 52.3  | 47.7      |
> | ✔              | ✗                       | 58.3  | 18.2 | 54.2  | 48.1      |
> | ✗              | ✔                       | 58.7  | 18.4 | 52.6  | 48.3      |
> | ✔              | ✔                       | 59.0  | 22.3 | 54.1  | 49.3      |
>
> **Table 3:** *Ablation study of each component in Qwen2.5 (14B → 1.5B). The baseline is on-policy JSD with a β=0.9 model (except all components).*
>
> We did not observe such redundancy in our ablation study.
> Each component improves different aspects of reasoning, and the highest gains are achieved when they are *combined*.
>
> Table 3 shows an ablation study comparing only two components: direct distillation with Entropy Gating (EG for short), and selective entropy minimization $\mathcal{L}_{\text{EM}}$, improve the model in distinctive ways.
>
> - **Effect of Entropy Gating alone (Row 1 $\rightarrow$ Row 2).**
>     Applying only EG produces small yet consistent gains across GSM8K (57.9 $\rightarrow$ 58.3), IFEVal (52.3 $\rightarrow$ 54.2), and MMLU-STEM (47.7 $\rightarrow$ 48.1). However, the improvement on MATH remains marginal (18.1 $\rightarrow$ 18.2). This shows that $\sigma_{\tau}(\cdot)$ primarily reduces distribution mismatch but does *not* provide strong reasoning supervision.
>
> - **Effect of $\mathcal{L}_{\text{EM}}$ alone (Row 1 $\rightarrow$ Row 3).** Using only $L_\text{EM}$ yields different gains: GSM8K and MATH improve to (57.9$\rightarrow$58.7) and (18.1$\rightarrow$18.4), respectively. Results show that the shifts indicate that $\mathcal{L}_{\text{EM}}$ directly supervises evidence alignment, unlike EG.
>
> - **Synergistic effect when combined (Row 1 $\rightarrow$ Row 4).** When EG and $\mathcal{L}_{\text{EM}}$ are used together, the model achieves a substantially higher MATH score (18.1 $\rightarrow$ 22.3), far exceeding the gains from either component alone. This *non-additive jump* demonstrates that the two modules address distinct error modes:
>     If the two losses were redundant, such a large improvement would not arise.
>
> From the results, we observe that neither component alone suffices to achieve the best performance. This trend is observed over diverse domains: GSM8K, MATH, IFval, and MMLU-STEM. In conclusion, the ablation results show *synergistic, not redundant* contributions.
>
> ---
>
> ***Q3: What steps are in place to detect or mitigate biases propagated from teacher to student, considering that only a subset of tokens is distilled but on the student’s own output?***
>
> Currently, our method has no additional steps for detecting or mitigating the teachers' biases. However, we argue that **our method is more robust to such biases** than most of existing methods as follows.
>
> The main objective of KD is to align the student’s distribution with that of the teacher. This inevitably entails the risk of inheriting the teacher's biases. Conventional KD methods, e.g., Supervised KD [1], GKD [2], etc., force the student to be aligned with the teacher's distribution over all the tokens. However, our **indirect and selective** approach mitigates the transfer of teacher's bias:
>
>
> - **Indirect Knowledge Transfer:** In our indirect distillation, the student proposes its own top tokens, and learns from the indirect preference signals on those tokens from the teacher. Such a weak and indirect form of knowledge transfer from the teacher mitigates the risk of learning teachers' biases.
>
> - **Selective Knowledge Transfer:** In our method, the knowledge transfer is mostly limited to high-confidence (low-entropy) tokens by the teacher to low-confidence tokens by the student. Compared to existing KD methods that distill the entire tokens, our selective transfer reduces the risk of overfitting the teacher's potential hallucinations or ambiguous biases.
>
> - **Autonomous Exploration:** By adopting the on-policy KD, the student learns from its own generated rollouts rather than fixed teacher references. This allows the student to actively navigate the output space, establishing its own reasoning paths independent of the teacher's specific patterns.
>
> **References**
>
> [1] Hinton, Geoffrey, Oriol Vinyals, and Jeff Dean. "Distilling the knowledge in a neural network." arXiv preprint arXiv:1503.02531 (2015).
>
> [2] Agarwal, Rishabh, et al. "On-policy distillation of language models: Learning from self-generated mistakes." The twelfth international conference on learning representations. 2024.

---

### Official Review · Reviewer_BR6w · 2025-10-31

**Soundness:** 3
**Presentation:** 3
**Contribution:** 2
**Rating:** 4
**Confidence:** 4

**Summary:**

The paper proposes Token-Selective Dual Knowledge Distillation (TSD-KD), a framework designed to efficiently transfer the reasoning abilities of a large teacher model to a smaller student model, aiming to reduce the cost of Chain-of-Thoughts (CoT) generation. Adopting a student-centric, on-policy distillation paradigm, TSD-KD applies supervision only to the most critical or uncertain tokens during the reasoning process, thereby avoiding the distribution mismatch and overwhelming issues associated with traditional Teacher-Forcing. The method integrates three key components, all guided by token selection: Indirect Distillation (teacher acts as a preference ranker for student candidates), Direct Distillation (applying GKD loss to tokens with a large uncertainty gap—student uncertain but teacher certain), and Entropy Regularization (selectively minimizing student entropy on critical tokens).

**Strengths:**

1) TSD-KD achieves State-of-the-Art performance across 10 challenging reasoning benchmarks. Experimental results demonstrate its significant superiority over existing baseline methods across multiple tasks.
2)  The student model, after training, even surpasses its teacher model on some reasoning tasks (with improvements up to 20.3%). This result strongly suggests the framework is not merely imitative but effectively promotes the student model in building its own, more generalizable reasoning logic.

**Weaknesses:**

1) The core insight of the paper—that "high-entropy/uncertain tokens are critical branching points in reasoning" and should be targeted for selective supervision—is not an original discovery. This phenomenon, which guides the model learning process, has been well-established in antecedent works (such as the RL-based methods by Wang et al. (2025) and Lei et al. (2025)). Therefore, the paper's contribution lies primarily in the engineering application and integration of this existing principle into the knowledge distillation domain for selective supervision, rather than a breakthrough in fundamental mechanism discovery or method innovation.
2) The TSD-KD methodology lacks deep theoretical innovation in distillation, being an effective combination of existing techniques and intuitive heuristic rules. Specifically, the Indirect Distillation employs the established Plackett-Luce (PL) model from RLHF, and Direct Distillation uses the known Generalized JSD (GKD) loss. While the "uncertainty gap" token selection mechanism is novel, it functions as an intuitive heuristic rule. Consequently, the paper's main contribution is the effective integration of these existing components, rather than the proposal of a new foundational distillation mechanism or a novel loss function.
3) The crucial length of the "Opener" for selective supervision is defined by an empirical hyperparameter: the c% accumulated entropy threshold (set to c=10% based on ablation studies). This fixed-ratio approach is a heuristic inherited from similar suggestions in other reinforcement learning works. The absence of a dynamic or adaptive mechanism that adjusts this threshold based on the specific complexity and depth of the reasoning task limits the theoretical generalizability of the method, as the optimal empirical value may vary significantly across different domains (e.g., mathematical vs. common-sense reasoning) and model architectures.

**Questions:**

1) Given that the insight "high-entropy/uncertain tokens contribute more" is highly similar to recent RL-based works (e.g., Wang et al. (2025) and Lei et al. (2025) as mentioned), how do the core innovative mechanisms of this paper (e.g., the uncertainty gap selection, the Dual Distillation design) demonstrate a theoretical or empirical advantage over the Token Importance mechanisms in the precursor works?
2) In the context of Knowledge Distillation, what specific advantages—such as increased data efficiency or stability—does selective supervision offer that cannot be achieved or are less efficient using traditional RL frameworks (i.e., penalizing/rewarding only critical tokens via sparse reward signals)?
3) Indirect Distillation is only applied during the Opener phase. How does the student model maintain reasoning consistency and quality during the subsequent unsupervised phases? If the student selects a path consistent with the teacher's preference during the Opener, to what extent does this restrict its ability to develop new, non-imitative reasoning logic "in its own words" in the subsequent steps?

---

> ### Author Response · Authors · 2025-11-24
> **Response Reviewer BR6w (1/5)**
>
> We sincerely appreciate the reviewer for valuable comments which are detailed, constructive, and to the point. We have addressed all of the concerns raised by the reviewer.
>
> ---
>
> ***W1: The core insight of the paper—that "high-entropy/uncertain tokens are critical branching points in reasoning" and should be targeted for selective supervision—is not an original discovery. This phenomenon, which guides the model learning process, has been well-established in antecedent works (such as the RL-based methods by Wang et al. (2025) and Lei et al. (2025)). Therefore, the paper's contribution lies primarily in the engineering application and integration of this existing principle into the knowledge distillation domain for selective supervision, rather than a breakthrough in fundamental mechanism discovery or method innovation.***
>
> **1. The key to our method is Dual Distillation.** Our main contribution is not limited to using high-entropy tokens. We propose Indirect and Direct Distillation and their complementary integration. Its design principle is centered on being *student-centric*, which we propose as a new paradigm for knowledge distillation (KD).
>
> **2. Our token selection takes a variety of forms,** and it is not merely based on high-entropy tokens, as explained below. (In fact, a simple "high-entropy token distillation" may not work well. Please refer to experiments in the response to **Q1**.)
>
> - **Cumulative entropy.**  We discovered the early concentration of entropy in the reasoning passages (Fig. 1 in the paper). We not only provided its intuitive explanation, but also proposed to use **cumulative entropy**, i.e., a new concept of *Openers*, for Indirect Distillation.
>
> - **Relative entropy.** Our direct distillation is based on the **relative entropy** between the student and teacher models. Such relativity or uncertainty gaps are present for KD problems, but not in plain RL. In addition, we provide some theoretical analysis of direct distillation with the uncertainty gap: please refer to the response to **W2**.
>
> **3. We address problems central to KD.** Our method covers key topics for KD, e.g., **distribution shift**, **capacity gap**, **model confidence**. The topics lie beyond those for RL with selective supervision (simple "selective supervision" RL may not work well. Please see the response to **Q2**.). To our belief, our method is methodologically (dual distillation) and philosophically (student-centric) innovative and can contribute to the KD research community.

---

> ### Author Response · Authors · 2025-11-24
> **Response Reviewer BR6w (2/5)**
>
> ***W2: The TSD-KD methodology lacks deep theoretical innovation in distillation, being an effective combination of existing techniques and intuitive heuristic rules. Specifically, the Indirect Distillation employs the established Plackett-Luce (PL) model from RLHF, and Direct Distillation uses the known Generalized JSD (GKD) loss. While the "uncertainty gap" token selection mechanism is novel, it functions as an intuitive heuristic rule. Consequently, the paper's main contribution is the effective integration of these existing components, rather than the proposal of a new foundational distillation mechanism or a novel loss function.***
>
> **1. Indirect Distillation is a new concept.** We respectfully argue that our method is more than a simple application of PL model [1]. Most of academic fields involving (re-)ranking, e.g., information retrieval [2], use the PL model. We proposed a PL model for **teacher's re-ranking of candidate tokens proposed by student** for our indirect distillation. To our belief, this is a new concept in KD. Moreover, we provide a theoretical grounding that our token ranking is equivalent to ranking **sub-responses** (sub-sequence of passages) modeling partial reasoning traces (please see Proposition 1 in the paper). In conclusion, Indirect Distillation is both novel and theoretically grounded.
>
> **2. Theoretical Insights for Direct Distillation.** As you have mentioned, the "uncertainty gap" is a novel element in our Direct Distillation. We provide some theoretical insights on our approach as follows.
>
> **Claim: Token selection with uncertainty gap can scale the gradient of the student model during training, facilitating convergence.**
>
> Our claim is based on the gradient rescaling property of KD [3]. For mathematical tractability, we only analyze forward KL, and assume that only the forward KL is used in our direct distillation. Consider the teacher model $p$ and the student model $q$. The student model is an autoregressive LLM whose output is the softmax of token logits. Specifically, let $q_i$ denote the confidence in token $i$ given by $q_i=\frac{\exp(z_i)}{\sum_j \exp(z_j)}$ where $z_i$ denotes the logit of token $i$. Consider the conventional (forward) KD: we minimize the loss
>
> $L=\mathcal D_{KL}(p\|q) $
>
> If we take the gradient of $L$ with respect to logit $z_i$:
>
> $\frac{dL}{dz_i} = p_i - q_i$
>
> This gradient is the *feedback* to logit $z_i$. Now suppose that token $i$ is the *ground truth* for a reasoning task. In reality, there rarely is a ground truth token for LLMs. However, since we consider *reasoning tasks*, there is a highly likely token leading to the correct answer. For example, the next predicted token for **The answer to 5+3 is** is **8**. We will assume that *Oracle* exists and provides the one-hot label for the "best token" at each reasoning step. We will simply refer to the best token labeled by Oracle as the ground truth. Equation (1) implies that, if the teacher $p_i$ has a higher confidence than the student $q_i$ $(p_i>q_i)$, then the feedback is positive. In that case, training will increase logit $z_i$ of the ground truth, which is desirable for training. This can speed up the model convergence [4].
>
> Moreover, if the student's confidence is lower (e.g., in the beginning of training), the gradient is relatively stronger. Specifically, consider two students $q$ and $q'$. Suppose $q$ has lower confidence on ground truth token $i$ than $q'$, i.e., $q_i < q_i'$. One can consider $q$ as the student during the early stage of training, and $q'$ as the same model at the later stage. We call $q$ a *weaker* student than $q'$. Consider the ratio of the gradients of $q$ and $q'$ of the logit of the $i$-th token:
>
> $$
> \nabla_i := \frac{dL/dz_i}{dL'/dz_i} = \frac{p_i - q_i}{p_i - q'_i} > 1
> $$
>
> Thus, $\nabla_i>1$ means that the weaker student or the student in the early training stage achieves an additional multiplicative gain on the gradient. $\nabla_i$ is called the *rescaling factor* of a gradient. The higher rescaling factor can accelerate the training of the student.
>
> (continue)

---

> ### Author Response · Authors · 2025-11-24
> **Response Reviewer BR6w (3/5)**
>
> Next, we consider the token-selective KD proposed in the paper. We consider "soft selection" based on entropy and sigmoid gating function as in Eq. (10) in the paper. The distillation objective has the following form (only forward KL is used):
>
> $$
> L_{\text{TS}} = \sigma(H_i(q)-H_i(p)) \cdot \mathcal D_{KL}(p\|q)
> $$
>
> where $H_i(p)$ is the entropy of token $i$ evaluated by $p$, and the sigmoid function $\sigma(\cdot)$ enables the "soft selection" based on the entropy difference. Its gradient is given by
>
> $$
> \frac{dL_{\text{TS}}}{dz_i}
> = \sigma(H_i(q)-H_i(p))[p_i - q_i] \tag{2}
> $$
>
> Note that the $\sigma(\cdot)$ term is simply a weight and is computed with no gradient flow to $p$ and $q$ in $\sigma(\cdot)$ (we apply stop-gradient). We can also define the rescaling factor with respect to student models $q$ and $q'$ ($q$ is the weaker student):
>
> $$
> \nabla_i^\text{TS} :=
> \frac{\sigma(H_i(q)-H_i(p))(p_i-q_i)}
> {\sigma(H_i(q')-H_i(p))(p_i-q_i')} \tag{3}
> $$
>
> From (3), we have that
>
> $$
> \nabla_i^{\textrm{TS}}=\underbrace{
> \frac{\sigma\left(H_i(q)-H_i(p)\right)}
> {\sigma\left(H_i(q')-H_i(p)\right)}
> }_{C}
> \nabla_i
> $$
>
>
> Thus, the token selection has a higher rescaling factor than the conventional KD if \(C>1\). We have that
>
> $$
> C =\frac{\sigma(H_i(q)-H_i(p))}
> {\sigma(H_i(q')-H_i(p))}
> =\frac{\exp(-H_i(p)) + \exp(-H_i(q'))}
> {\exp(-H_i(p)) + \exp(-H_i(q))} \tag{4}
> $$
>
>
> If the weaker student $q$ is *less* confident than $q'$ in token $i$, it has *higher* uncertainty. Thus, $H_i(q) > H_i(q')$, which implies that $C>1$. Conversely, if the confidence of the student is higher, $C<1$. Thus, the student will perceive $C>1$ during the early stage of training, while it will perceive $C<1$ during the later stage. This leads to two effects on the student models at different stages of training.
>
> **(1) Reinforced gradient.** In the early stage of training, the student model is relatively weak and perceives $C$ as $C>1$. This means that the scaling factor under token selection is stronger than the conventional KD. Thus, token selection can **reinforce** the gradient towards the "best" token. Such a reinforced gradient helps establish logical abilities in the early stage, facilitating the convergence of training.
>
> **(2) Label smoothing.** In the later stage of the training, the student is well-trained and is likely to have basic reasoning capabilities. The student will perceive $C$ as $C<1$ compared to the earlier version of itself. This means that the gradient is perceived weaker than the conventional KD. Thus, the student can *diversify* its reasoning process based on already established knowledge. Since the gradient to "best" answer has weakened, we can liken the approach to **label smoothing** [5]. It allows more exploration of parameter space for reasoning, which is consistent with our student-centric approach.
>
> **Action:** We will add the analysis to Appendix A.8 (due to space limitation). Please see Appendix A.8 of the revised paper.
>
> **References**
>
> [1] Plackett, Robin L. "The analysis of permutations." Journal of the Royal Statistical Society Series C: Applied Statistics 24.2 (1975): 193-202.
>
> [2] Xia, Fen, et al. "Listwise approach to learning to rank: theory and algorithm." Proceedings of the 25th international conference on Machine learning. 2008.
>
> [3] Tang, Jiaxi, et al. "Understanding and improving knowledge distillation." arXiv preprint arXiv:2002.03532 (2020).
>
> [4] Yoshua Bengio, Jérôme Louradour, Ronan Collobert, and Jason Weston. Curriculum learning. In
> Proceedings of the 26th annual international conference on machine learning, pp. 41–48. ACM,
> 2009.
>
> [5] Müller, Rafael, Simon Kornblith, and Geoffrey E. Hinton. "When does label smoothing help?." Advances in neural information processing systems 32 (2019).

---

> ### Author Response · Authors · 2025-11-24
> **Response Reviewer BR6w (4/5)**
>
> ***W3: 1. The crucial length of the "Opener" for selective supervision is defined by an empirical hyperparameter: the c\% accumulated entropy threshold (set to c=10\% based on ablation studies). 2. This fixed-ratio approach is a heuristic inherited from similar suggestions in other reinforcement learning works. The absence of a dynamic or adaptive mechanism that adjusts this threshold based on the specific complexity and depth of the reasoning task limits the theoretical generalizability of the method, as the optimal empirical value may vary significantly across different domains (e.g., mathematical vs. common-sense reasoning) and model architectures.***
>
> **1. Clarification.** We first clarify that $c=10$\% selection does **NOT** come from the existing work [1]. This is because we consider the **cumulative entropy** for Opener selection. Specifically, we first compute the cumulative distribution of the token entropy from the beginning of the sample. Then we select from the beginning of the passage to the 10\% point of the cumulative distribution as Opener.
>
> **2. Adaptive $c$.** Thank you for your insightful suggestion. Per your suggestion, we designed the Opener selection with adaptive $c$ as follows.
>
> **2-1. Adaptive method.**
>
> - For each sample, we estimate the sample "difficulty" based on the average uncertainty gap. This is consistent with our direct distillation with the uncertainty gap.
>
> $C=\sigma(\bar{H}(p_S) - \bar{H}(p_T))$
>
> where $p_S$, $p_T$ denote the student and teacher model, $\bar{H}$ is the average entropy of each generated sample.
>
> - The selection ratio $c$ of Opener is determined by the quantile interval of the estimated value of $C$. We take a moving average of $C$ over the window of 5 samples for the estimation of $C$.
>
> $c%$ =
>
> 5% for C ∈ [0.00, 0.25),
>
> 10% for C ∈ [0.25, 0.50),
>
> 15% for C ∈ [0.50, 0.75),
>
> 20% for C ∈ [0.75, 1.00]
>
> Intuitively, larger uncertainty gap $C$ implies that the current data is difficult for the student. Thus, we take a larger cumulative percentile (larger $c$) for supervision.
>
> | Method                 | GSM8K    | GSM-Plus | MATH      | MBPP     | IFEval   | MMLU-STEM |
> | ---------------------- | -------- | -------- | --------- | -------- | -------- | --------- |
> | Qwen2.5-14B (Teacher)  | 80.3     | 59.7     | 21.7      | 78.9     | 85.9     | 70.5      |
> | Qwen2.5-1.5B (Student) | 57.1     | 38.8     | 16.9      | 38.4     | 53.1     | 49.5      |
> | Sequence-Level KD      | 56.5     | 38.2     | 16.5      | 40.6     | 53.7     | 48.9      |
> | Supervised-KD          | 56.3     | 37.8     | 17.9      | 41.0     | 51.4     | 48.4      |
> | DistiLLM               | 57.2     | 38.7     | 18.6      | 42.1     | 52.5     | 47.8      |
> | MiniLLM                | 57.7     | 39.7     | 17.8      | **42.2** | 54.7     | 48.2      |
> | GKD (β=0.9)            | 57.9     | 39.9     | 18.1      | 41.8     | 52.3     | 47.7      |
> | TSD-KD (adaptive c)    | 59.3     | 39.8     | **26.5*** | 39.9     | **55.8** | **50.4**  |
> | TSD-KD                 | **60.1** | **40.5** | 26.1*     | 42.1     | 55.2     | 50.0      |
>
> Table 1.
>
> | Method                 | MMLU-Pro-Math | SciQ      | BBH      | MuSR     |
> | ---------------------- | ------------- | --------- | -------- | -------- |
> | Qwen2.5-14B (Teacher)  | 77.6          | 86.4      | 60.8     | 60.8     |
> | Qwen2.5-1.5B (Student) | 34.4          | 85.5      | 36.5     | 38.7     |
> | Sequence-Level KD      | 32.6          | 87.9      | 36.6     | 36.6     |
> | Supervised-KD          | 35.3          | 86.6      | 36.4     | 37.9     |
> | DistiLLM               | 35.2          | 85.7      | 36.5     | 38.7     |
> | MiniLLM                | 36.2          | 84.1      | 36.0     | 38.3     |
> | GKD (β=0.9)            | 35.7          | 86.6      | 36.2     | 38.5     |
> | TSD-KD (adaptive c)    | **38.5**      | **93.8*** | 40.0     | 39.4     |
> | TSD-KD                 | 36.9          | 93.0*     | **40.2** | **39.6** |
>
> Table 2.
>
> We conducted experiments with adaptive-$c$. Table 1,2 shows the results. Overall, adaptive-$c$ achieves a performance similar to the default (fixed-$c$). Interestingly, adaptive-$c$ perform better than fixed-$c$ on **MATH** and **SciQ** which are challenging reasoning benchmarks. Thus, the proposed adaptive-$c$ is good at adapting to the difficulty of distillation samples. Since fixed-$c$ and adaptive-$c$ perform similarly, we will leave them as two versions of our method.
>
> **Action:** We will add the results on the adaptive-$c$ method to the revised paper.
>
> **References**
>
> [1] Wang, Shenzhi, et al. "Beyond the 80/20 rule: High-entropy minority tokens drive effective reinforcement learning for llm reasoning." arXiv preprint arXiv:2506.01939 (2025).

---

> ### Author Response · Authors · 2025-11-24
> **Response Reviewer BR6w (5/5)**
>
> ***Q1: Given that the insight "high-entropy/uncertain tokens contribute more" is highly similar to recent RL-based works (e.g., Wang et al. (2025) and Lei et al. (2025) as mentioned), how do the core innovative mechanisms of this paper (e.g., the uncertainty gap selection, the Dual Distillation design) demonstrate a theoretical or empirical advantage over the Token Importance mechanisms in the precursor works?***
>
> A. This question has been answered mostly in the response to **W1**, so please refer to the response. In addition, we provide empirical evidence that simple "high-entropy token" distillation perform significantly worse than our method.
>
> Table 3 shows the comparison between the baseline and our method. For the baseline method:
>
> - Only direct distillation, or GKD, is used.
> - Only top-10\% entropy tokens are distilled.
>
> | Method KD            | GSM8K    | GSM Plus | MATH     | MBPP     | IFEVAL   | MMLU STEM |
> | -------------------- | -------- | -------- | -------- | -------- | -------- | --------- |
> | GKD w/ 10% selection | 58.2     | 38.2     | 18.5     | 40.4     | 50.1     | 47.9      |
> | Ours                 | **60.1** | **40.5** | **26.1** | **42.1** | **55.2** | **50.5**  |
>
> Table 3.
>
> We observe that our method significantly outperforms the baseline with simple selection. The results show the importance of capturing issues of knowledge distillation, e.g., distribution shift, capacity gap, confidence, etc., on top of token selection.
>
> ---
>
> ***Q2: In the context of Knowledge Distillation, what specific advantages—such as increased data efficiency or stability—does selective supervision offer that cannot be achieved or are less efficient using traditional RL frameworks (i.e., penalizing/rewarding only critical tokens via sparse reward signals)?***
>
> We examine the **data efficiency**. We compare RL with token selection using GRPO [1] and our proposed KD. We extracted 1k samples for training from [2], which contain mathematical questions and answers. We set the size of rollout $N=2$. Then, we evaluate several math benchmarks in Table 4. Overall, our method outperforms the RL baseline. Since both methods are trained on data with identical sample sizes, the result shows that our KD approach is superior in terms of data efficiency. The drawback of RL training is that it is very resource-intensive. This is the main reason we tested with a relatively small sample size and rollout.
>
> | Method KD | GSM8K     | GSM Plus  | IFEVAL    | MATH      | MMLU STEM |
> | --------- | --------- | --------- | --------- | --------- | --------- |
> | GRPO      | **56.85** | 38.80     | 52.84     | 17.10     | 47.6      |
> | Ours      | 56.56     | **39.39** | **53.00** | **20.16** | **48.2**  |
>
> Table 4.
>
> The results again show the importance of addressing key issues of knowledge distillation, e.g., **distribution shift**, **capacity gap**, **confidence**. We proposed Indirect Distillation to tackle **distribution shift**. We utilized the relative gap of entropy to leverage **capacity gap**. Finally we proposed entropy regularization to boost the student's **confidence**.
> These techniques are valid in the context of KD but are beyond that of conventional RL.
>
> **References**
>
> [1] Shao, Zhihong, et al. "Deepseekmath: Pushing the limits of mathematical reasoning in open language models." arXiv preprint arXiv:2402.03300 (2024).
>
> [2] https://huggingface.co/datasets/BytedTsinghua-SIA/DAPO-Math-17k
>
> ---
>
> ***Q3: Indirect Distillation is only applied during the Opener phase. How does the student model maintain reasoning consistency and quality during the subsequent unsupervised phases? If the student selects a path consistent with the teacher's preference during the Opener, to what extent does this restrict its ability to develop new, non-imitative reasoning logic "in its own words" in the subsequent steps?***
>
> Indirect Distillation, although it has a weak form, is still a type of distillation. The distillation of too many tokens may have a detrimental effect. For example, if we set Opener to be the entire sample tokens ($c=100$\%), the performance degrades: see Fig. 3 in the paper. However, as the reviewer noted, this leads to the problem that the tokens after Opener remain unsupervised.
>
> This is the reason why we use Direct Distillation. In Indirect Distillation, the tokens are selected in a contiguous manner. Specifically, it selects all the tokens up to a point where the cumulative entropy reaches $c\%$. In contrast, Direct Distillation can be performed on any token position in the sample, as long as that token has a high uncertainty gap. Thus, Direct Distillation can cover "difficult tokens" that appear later in the sample that were not supervised by Indirect Distillation. This is the main motivation of the dual distillation: **Indirect and Direct Distillation work in a complementary manner**. This dual approach is key to maintaining consistency and quality of reasoning.

---

### Official Review · Reviewer_jW9T · 2025-11-07

**Soundness:** 3
**Presentation:** 3
**Contribution:** 3
**Rating:** 6
**Confidence:** 4

**Summary:**

This paper proposes Token-Selective Dual Knowledge Distillation (TSD-KD), a framework for transferring large language model reasoning abilities to smaller models. The method is designed to provide targeted supervision, focusing on high-uncertainty tokens to mitigate issues with existing KD approaches. Three key components are listed:

1) Token-Selective Indirect Distillation: The teacher provides preference rankings over the student's top-k generated tokens in the initial sequence of reasoning (the "opener"), utilizing a Plackett-Luce model.
2) Token-Selective Direct Distillation: A JSD-based distribution matching loss is applied only to tokens where the student's uncertainty (entropy) significantly exceeds the teacher's confidence (the "uncertainty gap").
3) Token-Selective Entropy Regularization: The entropy of the student's top 10% most uncertain tokens is minimized.

**Strengths:**

1. The core idea of Token-Selective Direct Distillation is well motivated.
2. Authors provide comprehensive ablation study in demonstrating effects of each component and showed strong empirical results over baselines.

**Weaknesses:**

W1: Hyperparameter Sensitivity: The framework relies on an extremely sensitive set of hyperparameters ($c$, $k$, $s$, $\beta$), as demonstrated by sharp performance drop-offs in the appendix analyses. This suggests the method is brittle and lacks practical generalizability. In the Table 1, authors also only report the performance from the best hyperparamter selections. I wonder how much this complex setup could transfer into new domains or tasks.

W2: Conflict Between On-Policy Learning and Entropy Minimization ($\mathcal{L}_{EM}$): The $\mathcal{L}_{EM}$ term, which minimizes entropy on the top 10% most uncertain tokens, fundamentally conflicts with the core on-policy principle of preserving and encouraging exploration. While selectivity is claimed as a mitigation, the paper does not analyze the true impact on the student's output diversity or rigorously justify that minimizing entropy is superior to simpler confidence maintenance.

W3: The paper provides insufficient analysis to attribute the performance at Token-Selective Indirect Distillation. It is unclear if the success of the indirect distillation is due to the preference ranking (teacher's subtle knowledge transfer) or simply the top-k token candidate proposal. Based on prior work, latter might be the bigger contribution. I believe authors should perform additional ablation experiments to justify that preference ranking is necessary.

W4: There is a very relevant paper Speculative KD (https://arxiv.org/abs/2410.11325). Authors should consider compare to or mention.

**Questions:**

At W3

---

> ### Author Response · Authors · 2025-11-26
> **Response to Reviewer jW9T (1/4)**
>
> We sincerely appreciate the reviewer for valuable comments which are detailed, constructive, and to the point. We have addressed all of the concerns raised by the reviewer.
>
> ---
>
> ***W1: Hyperparameter Sensitivity: The framework relies on an extremely sensitive set of hyperparameters c,s,k,b, as demonstrated by sharp performance drop-offs in the appendix analyses. This suggests the method is brittle and lacks practical generalizability. In the Table 1, authors also only report the performance from the best hyperparamter selections. I wonder how much this complex setup could transfer into new domains or tasks.***
>
> To begin with, we would like to make the following clarifications.
>
> - Two of four hyperparameters mentioned ($\beta$ and $s$) are inherited from the values in prior work, but are used **without change**. Although we provided performance curves on $\beta$ and $s$, those were for completeness's sake. The curves are shown to reconfirm the validity of the original settings of prior work. No change from the original hyperparameters actually show the **simplicity and robustness** of our method.
>
> -  In the remaining two ($c$ and $k$), we newly propose the **adaptive-$c$** method to set $c$. The value of $c$ is chosen adaptively based on sample difficulty.
>
> Next, we explain the choice of hyperparameters in detail.
>
> **1. JSD parameter $\beta$.** The value of $\beta$ determines the ratio of mixture of forward and reverse KL in JSD loss. The value $\beta=0.9$ was directly inherited from the original GKD [1] paper. We provided performance with varying $\beta$ in Fig.6 of the paper to re-confirm this original choice. In fact, $\beta$ being optimal at the original value (0.9) shows the **robustness** of our method, because we do not need to reconfigure the parameter.
>
> **2. Selection ratio $s$.** The choice of $s$ follows the standard 80/20 recipe introduced in [2] to measure the token importance. The original recommendation was $s=$10\%$\sim$ 20\%, and we chose $s=$10\%.
> We provided performance with varying $s$ in Fig.5 of the paper to re-confirm this original choice. Again, $s$ being optimal within the range of the original value shows the **robustness** of our method without the need for re-configuration.
>
> **3. Top-$k$ selection.**
>
> We performed only a minimal sanity search over a small set of reasonable values of $k$. This is typical for conventional top-k methods.
>
> **4. Cumulative entropy ratio $c$.** The parameter $c$ is an important hyperparameter that determines the number of tokens to be supervised for Indirect Distillation. We found that $c=$10\% optimal. In the revised paper, instead of fixed $c$, we consider the extension of making $c$ adaptive. Specifically, we adapt the distillation or supervision ratio $c$ to the difficulty of the sample.
>
> (continue)

---

> ### Author Response · Authors · 2025-11-26
> **Response to Reviewer jW9T (2/4)**
>
> **4-1. Adaptive method.**
>
> Below we summarize the adaptive-$c$ method.
>
> - For each sample, we estimate the sample "difficulty" based on the average uncertainty gap. This is consistent with our direct distillation with the uncertainty gap.
>
> $C = \sigma(\bar{H}(p_S) - \bar{H}(p_T))$
>
> where $p_S$, $p_T$ denote the student and teacher model, $\bar{H}$ is the average entropy of each generated sample.
>
> - The selection ratio $c$ of Opener is determined by the quantile interval of the estimated value of $C$. We take a moving average of $C$ over the window of 5 samples for the estimation of $C$.
>
> $c%$ =
>
> 5% for C ∈ [0.00, 0.25),
>
> 10% for C ∈ [0.25, 0.50),
>
> 15% for C ∈ [0.50, 0.75),
>
> 20% for C ∈ [0.75, 1.00]
>
> Intuitively, larger uncertainty gap $C$ implies that the current data is difficult for the student. Thus, we take a larger cumulative percentile (larger $c$) for supervision.
>
> **4-2. Results.** We conducted experiments with adaptive-$c$. Table 1,2 shows the results. Overall, adaptive-$c$ achieves a performance similar to the default (fixed-$c$). Interestingly, adaptive-$c$ perform better than fixed-$c$ on **MATH** and **SciQ** which are challenging reasoning benchmarks. Thus, the proposed adaptive-$c$ is good at adapting to the difficulty of distillation samples. Since fixed-$c$ and adaptive-$c$ perform similarly, we will leave them as two versions of our method.
>
> | Method                 | GSM8K    | GSM-Plus | MATH      | MBPP     | IFEval   | MMLU-STEM |
> | ---------------------- | -------- | -------- | --------- | -------- | -------- | --------- |
> | Qwen2.5-14B (Teacher)  | 80.3     | 59.7     | 21.7      | 78.9     | 85.9     | 70.5      |
> | Qwen2.5-1.5B (Student) | 57.1     | 38.8     | 16.9      | 38.4     | 53.1     | 49.5      |
> | Sequence-Level KD      | 56.5     | 38.2     | 16.5      | 40.6     | 53.7     | 48.9      |
> | Supervised-KD          | 56.3     | 37.8     | 17.9      | 41.0     | 51.4     | 48.4      |
> | DistiLLM               | 57.2     | 38.7     | 18.6      | 42.1     | 52.5     | 47.8      |
> | MiniLLM                | 57.7     | 39.7     | 17.8      | **42.2** | 54.7     | 48.2      |
> | GKD (β=0.9)            | 57.9     | 39.9     | 18.1      | 41.8     | 52.3     | 47.7      |
> | TSD-KD (adaptive c)    | 59.3     | 39.8     | **26.5*** | 39.9     | **55.8** | **50.4**  |
> | TSD-KD                 | **60.1** | **40.5** | 26.1*     | 42.1     | 55.2     | 50.0      |
>
> Table 1.
>
> | Method                 | MMLU-Pro-Math | SciQ      | BBH      | MuSR     |
> | ---------------------- | ------------- | --------- | -------- | -------- |
> | Qwen2.5-14B (Teacher)  | 77.6          | 86.4      | 60.8     | 60.8     |
> | Qwen2.5-1.5B (Student) | 34.4          | 85.5      | 36.5     | 38.7     |
> | Sequence-Level KD      | 32.6          | 87.9      | 36.6     | 36.6     |
> | Supervised-KD          | 35.3          | 86.6      | 36.4     | 37.9     |
> | DistiLLM               | 35.2          | 85.7      | 36.5     | 38.7     |
> | MiniLLM                | 36.2          | 84.1      | 36.0     | 38.3     |
> | GKD (β=0.9)            | 35.7          | 86.6      | 36.2     | 38.5     |
> | TSD-KD (adaptive c)    | **38.5**      | **93.8*** | 40.0     | 39.4     |
> | TSD-KD                 | 36.9          | 93.0*     | **40.2** | **39.6** |
>
> Table 2.
>
> **Action:** We will add the results on the adaptive-$c$ method to the revised paper.
>
> **References**
>
> [1] Agarwal, Rishabh, et al. "On-policy distillation of language models: Learning from self-generated mistakes." The twelfth international conference on learning representations. 2024.
>
> [2] Wang, Shenzhi, et al. "Beyond the 80/20 rule: High-entropy minority tokens drive effective reinforcement learning for llm reasoning." arXiv preprint arXiv:2506.01939 (2025).

---

> ### Author Response · Authors · 2025-11-26
> **Response to Reviewer jW9T (3/4)**
>
> ***W2: Conflict Between On-Policy Learning and Entropy Minimization $\mathcal{L}_{EM}$ term, which minimizes entropy on the top 10% most uncertain tokens, fundamentally conflicts with the core on-policy principle of preserving and encouraging exploration. While selectivity is claimed as a mitigation, the paper does not analyze the true impact on the student's output diversity or rigorously justify that minimizing entropy is superior to simpler confidence maintenance.***
>
> **(1) Selective and localized minimization is performed.**
> As the reviewer noted, entropy minimization is applied only to the top 10\% highest-entropy tokens that correspond to critical branching points; 90\% of tokens remain untouched. This is consistent with the two common principle of our model components (indirect distillation, direct distillation, entropy minimization):
>
> - *Encourage the student to freely generate less-difficult tokens.* Indirect or direct distillation is **not** applied to easy tokens. In entropy minimization, such less-difficult tokens consist of 90\% of all tokens. Thus, the impact on diversity is minimal.
>
> - *Focus the distillation on difficult tokens*. Indirect and direct distillation are focused on those tokens. In entropy minimization, the student is made more deterministic (confident) rather than explorative. This can improve the effectiveness of distilling challenging knowledge.
>
> Of course, the whole generation uses the on-policy KD. Thus, the high-level principle is to encourage the student's own generation; however, low-level details should capture the training efficiency of distillation.
>
>
> **(2) Entropy minimization outperforms confidence maintenance.**  The "confidence maintenance" mentioned by the reviewer corresponds to **not** using entropy minimization. Thus, their comparison can be found in the the **ablation study on entropy minimization.** As shown in Table 3 in the paper, adding the entropy loss $\mathcal{L}_{\mathrm{EM}}$ *always improves* performance.
>
> - With $\sigma_\tau$ fixed, adding only $\mathcal{L}_{\mathrm{EM}}$ improves MATH: 20.9 $\rightarrow$ 22.3, GSM8K: 58.6 $\rightarrow$ 59.0.
>
> - When all components are enabled, the model achieves the best scores across **all** benchmarks  (e.g., **26.1** MATH, **60.1** GSM8K).
>
> ---
>
> ***W3: The paper provides insufficient analysis to attribute the performance at Token-Selective Indirect Distillation. It is unclear if the success of the indirect distillation is due to the preference ranking (teacher's subtle knowledge transfer) or simply the top-$k$ token candidate proposal. Based on prior work, latter might be the bigger contribution. I believe authors should perform additional ablation experiments to justify that preference ranking is necessary.***
>
> We are sorry, but are slightly unsure whether we interpreted your question correctly.
> Top-$k$ tokens are proposed by the **student** itself, but not by the **teacher**. We will provide the responses to cover both cases.
>
> **Case 1: The question is on the teacher's top-$k$ token proposal.**
>
> In our indirect distillation, the teacher does not propose top-$k$ candidate tokens to the student. Instead, the student first produces its own top-$k$ candidates, and the teacher is only used to selectively refine these student-generated proposals. Thus, the teacher’s role is restricted to providing local corrective signals on the student’s existing token hypotheses.
>
> **Case 2: The question is on the student's top-$k$ token proposal.**
>
> In our indirect distillation, the top-$k$ proposal by the student and their ranking by the teacher constitute a single method. We are unsure how we can separate them.
>
> For example, if we summarize the ablation proposed by the reviewer, the method will consist of:
>
> - Student generates the top-$k$ proposal
> - Perform indirect distillation **without** the teacher's ranking on the student's top-$k$ proposal
>
> However, if the teacher does not provide ranking information on the top-$k$ proposal by the student, we hardly see other ways to pass the teacher's knowledge to the student. An alternative method is that (i) the student propose top-1 token to the teacher; (ii) the teacher provide its logit (confidence) on that token. However, this is the **direct** distillation (a forward KD), for which we provide a comparison in the paper. Please let us know if we misunderstood your question.

---

> ### Author Response · Authors · 2025-11-26
> **Response to Reviewer jW9T (4/4)**
>
> ***W4: There is a very relevant paper Speculative KD (https://arxiv.org/abs/2410.11325). Authors should consider compare to or mention.***
>
> Thank you for the suggestion. We will add Speculative KD as a baseline to our experiments. Below, we clarify the methodological differences and summarize the new comparison results.
>
> **Speculative KD:** The student first generates output tokens, after which the teacher may replace the student's low-quality proposals with teacher-sampled tokens on the fly. In other words, the teacher can inject high-probability tokens that were not originally within the student's top-$k$ candidates, effectively altering the candidate set from a teacher-centric perspective during training.
>
> **TSD-KD:** The proposed Token-Selective Indirect Distillation is student-centric: the student first generates its own top-$k$ candidates, and the teacher only provides selective preference signals (re-ranking / corrective scores) over this existing candidate set. Importantly, TSD-KD does not allow the teacher to replace or add tokens outside the student’s top-$k$; the teacher’s role is to order the student’s own hypotheses and provide local corrective feedback. This design (i) keeps the student’s hypothesis space fixed, (ii) focuses the signal on ranking/selection rather than candidate expansion, and (iii) avoids teacher-driven distribution shifts in the student’s on-policy samples.
>
> ***New experimental comparison.*** We added a direct empirical comparison to Speculative KD (Table 3,4) in our experimental settings. The table shows that our method outperforms Speculative KD across diverse evaluated benchmarks:
>
> | **Method**        | **GSM8K** | **GSM-Plus** | **MATH**  | **MBPP** | **IFEval** | **MMLU-STEM** |
> | ----------------- | --------- | ------------ | --------- | -------- | ---------- | ------------- |
> | Sequence-Level KD | 56.5      | 38.2         | 16.5      | 40.6     | 53.7       | 48.9          |
> | Supervised-KD     | 56.3      | 37.8         | 17.9      | 41.0     | 51.4       | 48.4          |
> | DistiLLM          | 57.2      | 38.7         | 18.6      | 42.1     | 52.5       | 47.8          |
> | MiniLLM           | 57.7      | 39.7         | 17.8      | **42.2** | 54.7       | 48.2          |
> | GKD (β=0.9)       | 57.9      | 39.9         | 18.1      | 41.8     | 52.3       | 47.7          |
> | Speculative KD    | 58.0      | 39.2         | 18.5      | 41.6     | 52.4       | 47.8          |
> | **TSD-KD**        | **60.1**  | **40.5**     | **26.1*** | 42.1     | **55.2**   | **50.0**      |
>
> Table 3.
>
> | **Method**        | **MMLU-Pro-Math** | **SciQ**  | **BBH**  | **MuSR** |
> | ----------------- | ----------------- | --------- | -------- | -------- |
> | Sequence-Level KD | 32.6              | 87.9      | 36.6     | 36.6     |
> | Supervised-KD     | 35.3              | 86.6      | 36.4     | 37.9     |
> | DistiLLM          | 35.2              | 85.7      | 36.5     | 38.7     |
> | MiniLLM           | 36.2              | 84.1      | 36.0     | 38.3     |
> | GKD (β=0.9)       | 35.7              | 86.6      | 36.2     | 38.5     |
> | Speculative KD    | 34.8              | 86.7      | 37.2     | 37.7     |
> | **TSD-KD**        | **36.9**          | **93.0*** | **40.2** | **39.6** |
>
> Table 4.
>
> **Summary.** Our method is algorithmically distinct from Speculative KD. Speculative KD replaces student proposals with teacher tokens (expanding the training candidate set). Our method keeps the student’s top-$k$ intact and applies selective, post-hoc preference ranking over those candidates. We have added direct comparisons (Table 3,4) showing that TSD-KD yields consistently larger gains than SKD on several reasoning benchmarks. Therefore, the improvements in our paper are not primarily due to mere exposure to wider or teacher-proposed candidates, but rather arise from the teacher’s subtle re-ranking signal applied to the student’s own hypotheses.
>
> **Action:** We will add Speculative KD [1] in the related work and add the results of Table 3,4 to the revision.
>
> [1] Wenda Xu, Rujun Han, Zifeng Wang, Long Le, Dhruv Madeka, Lei Li, William Yang Wang, Rishabh Agarwal, Chen-Yu Lee, & Tomas Pfister (2025). Speculative Knowledge Distillation: Bridging the Teacher-Student Gap Through Interleaved Sampling. In The Thirteenth International Conference on Learning Representations.

---

### Author Response · Authors · 2025-12-02
**General Response and Revision Summary**

Dear AC and Reviewers,

First of all, thank you for your service during this difficult time. Below, we provide a summary of score updates and rebuttals.

**1. Score Updates.**

**- Summary.**

- **Initial Scores:** 6, 6, 4, 4

- **Post-discussion Scores:** **6, 6, 6, 4**

**- Score Increase.**

**Reviewer DKho (4 $\rightarrow$ 6):**
On Nov 25, Reviewer DKho explicitly stated:
*“Thanks for addressing my concerns. I have increased my score accordingly.”* The score increase followed our newly added experiments on Qwen3 and the theoretical derivation, which resolved the reviewer's concerns about generalizability and theoretical clarity.

**- Considerations.**

Due to the shortened rebuttal period, we were unable to receive responses from **three out of four** reviewers. We would appreciate if the AC could consider this difficulty and examine the detailed rebuttal we provided for the comments.

**2. Summary of Strengths.**

We thank the reviewers for encouraging comments including:

- **Novelty and Theoretical Motivation:**

"The use of student-generated candidates ... and selective, entropy-based token gating ... is thoughtfully motivated and distinguishes the framework from prior 'teacher-forcing' approaches."

"The focus on letting the student 'explain in its own words' resonates with cognitive insights."

"The mathematical formulation is transparent, the underlying assumptions are stated, and the algorithmic components are described with appropriate rigor."

- **Comprehensive Framework Design:**

"Complete and sound framework combining distillation with entropy regularization forms a coherent pipeline."

"The explicit combination of indirect and direct knowledge distillation ... is well-positioned to address known weaknesses of pure distribution-matching."

- **Strong Empirical Results and Generalizability:**

"TSD-KD achieves State-of-the-Art performance across 10 challenging reasoning benchmarks."

"Comprehensive experiments across two model families demonstrate generalizability."

- **Efficacy of Student Reasoning:**

"The student model, after training, even surpasses its teacher model on some reasoning tasks (w/ improvements up to 20.3%)."

"This result strongly suggests the framework is not merely imitative but effectively promotes the student model in building its own, more generalizable reasoning logic."

**3. Major Concerns Addressed During Rebuttal.** All questions raised by the reviewers are addressed in our rebuttal. The following is a summary of revisions and rebuttals provided to address the reviewers' concerns.

- **State-of-the-Art Performance on Latest Models (DKho, jW9T)**
We added extensive experiments on the **Qwen3 family (8B → 1.7B)**. **Result:** Our method achieved state-of-the-art performance on most of the benchmarks. In particular, **the student outperformed the teacher by +5.6** on MATH.

- **Comparison with a strong baseline (jW9T)**
We added a full comparative experiment against a strong baseline: Speculative KD [1].  **Result:** TSD-KD consistently outperformed Speculative KD on all benchmarks.

- **Distinction from RL methods (BR6w)** The question was on how our work is distinguished from the prior RL method with selective supervision. We showed that the key is dual distillation: indirect and direct distillation, a new concept beyond selective supervision, along with their theoretical insights. We also provided comparisons with simple high-entropy token distillation (response to **Q1**) and RL with selective supervision (response to **Q2**). In both experiments, our method outperformed them with significant gaps.

- **Theoretical Grounding (BR6w, DKho)**
We provided a theoretical analysis in Appendix A.8 showing how uncertainty-gap-based selection induces a gradient rescaling effect. This provides the theoretical justification on the training stability of our method.


- **Hyperparameters and its adaptive extensions (jW9T, BR6w)** Some of the hyperparameters are inherited from prior work, but are reused without change. This actually shows the simplicity and robustness of our method. In addition, we provided the adaptive extension of another hyperparameter.

- **Robustness of Indirect Distillation (9rZk)**: We performed experiments on a weak student (Qwen2.5-0.5B). Our method with indirect distillation outperformed the off-policy method and on-policy method with direct distillation.

- **Statistical significance tests (9rZk)**: We demonstrate that the superiority of our method over existing baselines is statistically significant ($p < 0.05$) by Bootstrap resampling.

**4. Conclusion.**

We thank the reviewers for their constructive comments to improve our work. We also thank the new AC for your extra service. We hope you could consider our difficulty in reviewer interactions during the rebuttal period cut short.

---

[1] Wenda Xu et al. Speculative Knowledge Distillation: Bridging the Teacher-Student Gap Through Interleaved Sampling. ICLR 2025

---

### Meta-Review · Area_Chair_frjv · 2026-01-07

**Summary:**

The reviewers' primary concerns initially centered on the practical generalizability and theoretical depth of the proposed Token-Selective Dual Knowledge Distillation (TSD-KD) framework. Specifically, reviewers **jW9T** and **BR6w** highlighted a high sensitivity to hyperparameters ($c, k, s, \beta$) and questioned whether the "student-centric" approach was truly novel compared to existing reinforcement learning (RL) methods that also target high-entropy tokens. Reviewer **DKho** raised significant concerns regarding the use of relatively outdated models (Qwen2.5 and Gemma2) and a lack of theoretical justification for the entropy-based selection. Furthermore, reviewer **9rZk** questioned the robustness of indirect distillation for weaker student models and the statistical significance of the reported gains. Despite these initial reservations, the reviewers recognized the framework's strong empirical performance, particularly the notable result where student models occasionally outperformed their teachers in reasoning tasks.

**Reviewer Concerns:**

The authors effectively addressed the majority of the reviewers' technical concerns during the rebuttal. To mitigate hyperparameter sensitivity and the lack of dynamic thresholds (a concern for **jW9T** and **BR6w**), the authors introduced an "adaptive-" method that adjusts supervision based on sample difficulty, which maintained or improved performance on challenging benchmarks like MATH. The concern regarding outdated models (**DKho**) was resolved by adding extensive experiments on the Qwen3 family, where TSD-KD achieved state-of-the-art results. **DKho**'s and **BR6w**'s request for theoretical grounding was met with a new derivation in Appendix A.8, demonstrating how uncertainty-gap-based selection induces a gradient rescaling effect that stabilizes training. Additionally, the authors provided a direct comparison with "Speculative KD" to satisfy **jW9T** and conducted statistical significance tests (Bootstrap resampling) to satisfy **9rZk**. The concern from **BR6w** regarding the distinction from RL was addressed by demonstrating that dual distillation (indirect and direct) is algorithmically distinct and empirically superior to simple high-entropy token selection or pure RL baselines.

**Reviewer Scores:**

Due to the shortened rebuttal period, only **Reviewer DKho** updated their score, increasing it from **4 to 6** after the authors provided the Qwen3 experiments and theoretical derivations. Had they been able to participate fully in the discussion, it is likely that **Reviewer BR6w** would have also increased their score from **4 to 6**, as their primary requests for an adaptive mechanism and theoretical insight were directly addressed. **Reviewer jW9T** and **Reviewer 9rZk** (both initial **6**) likely would have maintained their scores or slightly increased them (to 7), as the authors provided the requested comparative baselines and statistical rigor. The overall consensus shifted toward a positive view of the paper’s empirical strength and its refined theoretical/methodological framing.

From the view of AC, this paper presents a novel and effective "student-centric" distillation paradigm that consistently outperforms existing methods across ten benchmarks. The authors’ responsiveness in providing new theoretical analysis, adaptive mechanisms, and updated model evaluations has significantly strengthened the submission.

---

### Decision · Program_Chairs · 2026-01-26

Accept (Poster)